# Attributing ozone and its precursors to land transport emissions in Europe and Germany

Mariano Mertens[1], Astrid Kerkweg[2,a], Volker Grewe[1,3], Patrick Jöckel[1], and Robert Sausen[1]

[1]Deutsches Zentrum für Luft- und Raumfahrt, Institut für Physik der Atmosphäre, Oberpfaffenhofen, Germany
[2]Institut für Geowissenschaften und Meteorologie, Rheinische Friedrich-Wilhelms-Universität Bonn, Germany
[3]Delft University of Technology, Aerospace Engineering, Section Aircraft Noise and Climate Effects, Delft, the Netherlands
[a]now at: IEK-8, Forschungszentrum Jülich, Jülich, Germany

**Correspondence:** Mariano Mertens (mariano.mertens@dlr.de)

**Abstract.** Land transport is an important emission source of nitrogen oxides, carbon monoxide and volatile organic compounds. The emissions of nitrogen oxides affect air quality directly. Further, all of these emissions serve as precursor for the formation of tropospheric ozone, thus leading to an indirect influence on air quality. In addition, ozone is radiatively active and its increase leads to a positive radiative forcing. Due to the strong non-linearity of the ozone chemistry, the contribution of emission sources
to ozone cannot be calculated or measured directly. Instead, atmospheric-chemistry models equipped with specific source attribution methods (e.g. tagging methods) are required. In this study we investigate the contribution of land transport emissions to ozone and ozone precursors using the MECO(n) model system. This model system couples a global and a regional chemistry climate model and is equipped with a tagging diagnostic. We investigate the combined effect of long range transported ozone and ozone which is produced by European emissions by applying the tagging diagnostic simultaneously and consistently on
the global and regional scale. We performed two simulations each covering three years with different anthropogenic emission inventories for Europe. We applied two regional refinements, i.e. one refinement covering Europe (50 km resolution) and one covering Germany (12 km resolution). The diagnosed absolute contributions of land transport emissions to reactive nitrogen ($NO_y$) near ground-level are in the range of 5 to 10 $nmol\,mol^{-1}$. This corresponds to relative contributions of 50 to 70 %. The largest absolute contributions appear around Paris, Southern England, Moscow, the Po Valley, and Western Germany.
The absolute contributions to carbon monoxide range from 30 $nmol\,mol^{-1}$ to more than 75 $nmol\,mol^{-1}$ near emission hotspots such as Paris or Moscow. The ozone which is attributed to land transport emissions shows a strong seasonal cycle with absolute contributions of 3 $nmol\,mol^{-1}$ during winter and 5 to 10 $nmol\,mol^{-1}$ during summer. This corresponds to relative contributions of 8 to 10 % during winter and up to 16 % during summer. The largest values during summer are confined to the Po Valley, while the contributions in Western Europe range from 12 to 14 %. Only during summer the ozone contributions are
slightly influenced by the anthropogenic emission inventory, but these differences are smaller than the range of the seasonal cycle of the contribution to land transport emissions. This cycle is caused by a complex interplay of seasonal cycles of other emissions (e.g. biogenic) and seasonal variations of the ozone regimes. In addition, our results suggest that during events with large ozone values the ozone contributions of land transport and biogenic emissions increase strongly. Here, the contribution of land transport emissions peak up to 28 %. Hence, our model results suggest that land transport emissions are an important
contributor during periods with large ozone values.

# 1 Introduction

Mobility plays a key role in everyday life, which involves the transport of goods and persons. Most of the transport processes rely on vehicles with combustion engines, which emit not only $CO_2$, but also many gaseous and particulate components, such as nitrogen oxides ($NO_x$), volatile organic compounds (VOCs), carbon monoxide (CO) or black carbon.

The transport sector with the largest emissions is the land transport sector (involving road traffic, inland navigation and trains). Even though the global emissions of many chemical species from the land transport sector decreased (e.g. Crippa et al., 2018), the emissions are still very large. For Europe and North America the emissions of $NO_x$ from road traffic have recently been the subject of public debate (e.g. Ehlers et al., 2016; Ntziachristos et al., 2016; Degraeuwe et al., 2017; Peitzmeier et al., 2017; Tanaka et al., 2018). $NO_x$ emissions influence the local air quality and lead to exceedances of the nitrogen dioxide ($NO_2$)

thresholds in many cities. Furthermore, $NO_x$ plays an important role for the tropospheric ozone chemistry and serves, together with CO and VOCs, as precursor for the formation of tropospheric ozone (e.g. Crutzen, 1974). Ozone is a strong oxidant and affects air quality (e.g. World Health Organization, 2003; Monks et al., 2015). Large ozone levels impact the vegetation and decrease crop yield rates (e.g. Fowler et al., 2009; Mauzerall et al., 2001; Teixeira et al., 2011). Furthermore, ozone is radiatively active and thus contributes to global warming (e.g. Stevenson et al., 2006; Myhre et al., 2013).

To quantify the influence of a specific emission source, such as land transport emissions on ozone, source apportionment methods are needed. Typically, two different methods are used for source apportionment. The first method is the *perturbation method*. In the perturbation method (also known as sensitivity analysis, brute-force, or zero-out) the results of two model simulations, one with all emissions and one with changed emissions, are compared. The second method is based on a labelling technique (known as tagging) to attribute specific pollutants, such as for instance ozone, to specific emission sources. Hereafter,

we refer to this method as *source attribution*.

As outlined in different studies (Wang et al., 2009; Grewe et al., 2010; Clappier et al., 2017) both methods answer different question because of their fundamentally different concepts. The perturbation method quantifies the change of ozone due to an emission change. In this method the sensitivity of ozone to this emission change is analysed based on a Taylor approximation (Grewe et al., 2010). In contrast, source attribution gives no information about the sensitivity of ozone to an emission change.

Instead, the share of ozone which is caused by the emissions of a specific emission source for a given state of the atmosphere is quantified. Therefore, we use hereafter the terms 'impact' for the results of the perturbation method and 'contribution' for results of source attribution.

The characteristics of impacts and contributions are listed in Table 1. By design, source attribution methods decompose the ozone budget completely into their respective contributions (this could be emission sectors, geographical regions, combinations

of this or other measures). Contributions calculated by source attribution are of interest for academic purpose to study the tropospheric ozone budget and to increase scientific understanding about factors determining ozone levels (e.g. Horowitz and Jacob, 1999; Lelieveld and Dentener, 2000; Meijer et al., 2000; Dunker et al., 2002; Grewe, 2004; Sudo and Akimoto, 2007; Dahlmann et al., 2011; Butler et al., 2018). Further, the knowledge about contributions can help the planning of mitigation options by finding the emission source, which contributes most to ozone (e.g. Kwok et al., 2015; Valverde et al., 2016; Pay et al.,

2019). Furthermore, the contributions are very valuable for assessing possible changes in the tropospheric ozone budget due to changes of emissions or climate. However, contributions provide no information about the sensitivity of ozone with respect to an emission change, such as the resulting ozone change, when emissions of a specific emission source become reduced or increased. The answers to such questions require the perturbation method, which quantifies the impact of an emissions change onto ozone. In contrast to the contributions, the effect of an emission reduction (and therefore the impact) can be measured. However, the results of the perturbation approach provide no information about how the effect of an emission reduction is altered by compensating effects of other emission sources (for instance an increase of ozone productivity of an unmitigated source). In order to assess such effects, perturbation and source attribution methods must be combined (see Mertens et al., 2018).

For a chemical specie that is controlled by linear processes only, the perturbation method and the source attribution methods lead to identical results. However, the ozone chemistry is strongly non-linear. Only for small perturbations around the base state (w.r.t. the chemical regime) the response of ozone to a small emission change can be considered as almost linear. Whether a response to an emission change is nearly linear depends on the chemical regime, and therefore the region and the considered time period. Thus, the perturbation approach does not allow for a complete ozone source attribution (e.g. Wild et al., 2012), because the impacts calculated for the different sectors do not sum up to 100 %. This leads to an underestimation of the contribution of specific emission sources to ozone, if these impacts are used for source attribution. As an example, Emmons et al. (2012) reported that tagged ozone is 2–4 times larger than the impact calculated by the perturbation approach. Even though the difference between impact and contribution is well known in the literature, the perturbation method is still widely used for ozone attribution studies, i.e. studies in which the contributions of emission sources to the ozone budget are analysed.

In the present study we want to investigate the share of land transport emissions to European ozone levels. Therefore, we choose a source attribution method to calculate the contributions of land transport emissions to ozone and ozone precursors. The effect of mitigation options of land transport emissions is not subject of this study. From the point of view of air quality planning this might be seen as an academic question, but as similar previous studies (e.g., Brandt et al., 2013; Karamchandani et al., 2017; Pay et al., 2019; Lupaşcu and Butler, 2019) our investigation improves the understanding of European ozone levels.

Many studies were performed which investigated the influence of land transport emissions to ozone on the global scale (e.g. Granier and Brasseur, 2003; Niemeier et al., 2006; Matthes et al., 2007; Hoor et al., 2009; Dahlmann et al., 2011; Mertens et al., 2018). All of them showed that land transport emissions impact ozone considerably on the global scale, especially in the northern hemisphere. These results of global models, however, give only very limited information on the contributions of the land transport (or other) emissions to ozone levels on the regional scale, especially as simulated ozone mixing ratios depend on the model resolution (e.g. Wild and Prather, 2006; Wild, 2007; Tie et al., 2010; Holmes et al., 2014; Markakis et al., 2015). Even though, land transport is, besides other anthropogenic emissions (e.g. Matthias et al., 2010; Tagaris et al., 2014; Aulinger et al., 2016; Yan et al., 2018) and biogenic emissions (e.g. Simpson, 1995; Solmon et al., 2004; Curci et al., 2009; Sartelet et al., 2012), an important source of ozone precursors in Europe, only few studies investigated the influence of European land transport emissions on ozone. Reis et al. (2000) investigated the impact of a projected change of road traffic emissions

from 1990 to 2010 on ground-level ozone in Europe, reporting a general decrease of ozone levels due to emission reductions. Similarly, Tagaris et al. (2015) quantified the impact of ten different emission sources on European ozone and PM2.5 levels using the CMAQ model for a specific period (July 2006). Tagaris et al. (2015) reported an impact of road transport emissions on the maximum 8-hour ozone mixing ratio of 10 % and more in Central Europe. Compared to this, Valverde et al. (2016) used

a source attribution method integrated in CMAQ (Kwok et al., 2015) to investigate the contributions of road traffic emissions of Madrid and Barcelona to ozone. They reported ozone contributions of 11 to 25 % for the Iberian Peninsula. Similarly, Karamchandani et al. (2017) applied the source attribution technique integrated in CAMx (Dunker et al., 2002) to calculate the contribution of eleven source categories on ozone concentrations for one summer and one winter month in 2010, focusing on 16 European cities. Generally, Karamchandani et al. (2017) reported contributions of 12 to 35 % of the road traffic sector on

the ozone levels in different cities. In accordance with other studies Karamchandani et al. (2017) showed that European ozone levels are strongly influenced by long range transport (e.g. Jonson et al., 2018; Pay et al., 2019). Despite the large importance of long range transport, all discussed studies applied the source attribution method in the regional model only. Ozone and ozone precursors which are advected towards Europe (i.e. significantly influenced by boundary conditions of the regional model) are not attributed to specific emission sources (or regions) but are attributed to the boundary conditions only.

Accordingly, all of the previous studies quantified only the contribution of European land transport emissions to the European ozone levels. In contrast, the present study provides a detailed assessment on the contribution of land transport emissions to ozone and ozone precursors ($NO_x$, CO) considering the combined effect of European and global emissions.

To include also the effects of long range transport in regional studies, a global-regional model chain is necessary, which includes a source attribution method in the global and the regional models. Such a model is the MECO(n) model system

(e.g. Kerkweg and Jöckel, 2012a, b; Hofmann et al., 2012; Mertens et al., 2016), which couples the global chemistry climate model EMAC (e.g Jöckel et al., 2010, 2016) at runtime to the regional chemistry model COSMO-CLM/MESSy (Kerkweg and Jöckel, 2012b). Two regional model refinements are applied, covering Europe and Germany with 50 km and 12 km resolution, respectively. The global model resolution is 300 km. The global and the regional models are equipped with the MESSy interface (Jöckel et al., 2005, 2010) and we apply the same tagging method (Grewe et al., 2017) for source attribution in the global and

the regional model. Compared to previous studies, this model system allows a contribution analysis from the global to the regional scale taking into account the effects of long range transport (Mertens et al., 2020).

Typically, the uncertainties of such source attribution studies are large. Reasons are:

- uncertainties in the models (e.g. chemical/physical parametrizations);

- uncertainties due to the choice of source attribution methods;

- uncertainties of the emissions inventories;

- seasonal variability of the contributions caused by meteorological conditions and seasonal cycles of emissions (e.g. stronger biogenic emissions and more active photochemistry during summer than winter);

– year to year variability of the contributions caused by meteorological conditions or large emissions of specific sources in specific years (for example yearly differences of biomass burning emissions);

To account for the uncertainties due to different emission inventories simulations with two different anthropogenic emission inventories were performed. To further account for the seasonal variability we investigate the contributions for winter and summer seasons. In addition, we consider always three simulation years to estimate the variability of the contributions between different years. The investigation of uncertainties caused by models and/or source attribution methods is beyond the scope of this study.

In our analysis we focus on mean and extreme (expressed as 95th percentile) contributions for the multi-year (2008 to 2010) seasonal average values of winter (December, January, February, hereafter DJF) and summer conditions (June, July, August, hereafter JJA). Our main priority is on the results of the European domain. However, as the model resolution can influence the results, we further investigate also results for the smaller domain covering Germany.

The manuscript is structured as follows. First, Section 2 contains a brief description of the model system, including an introduction to the applied tagging method, a description of the performed model simulations and the applied emission inventories, and a brief comparison of the simulated ozone concentrations with observations. Sections 3 and 4 discuss the contributions of land transport emissions to reactive nitrogen, carbon monoxide and ozone in Europe. Section 5 focuses on the contribution of reactive nitrogen for Germany only based on the finer resolved simulation results. Finally, the net ozone production over Europe and in particular the contributions of land transport emissions to the net ozone production are investigated in Section 6.

## 2 Description of the model system

In this study the MECO(n) model system is applied (Kerkweg and Jöckel, 2012b; Hofmann et al., 2012; Mertens et al., 2016; Kerkweg et al., 2018). This system couples on-line the global chemistry-climate model EMAC (Jöckel et al., 2006, 2010) with the regional scale chemistry-climate model COSMO-CLM/MESSy (Kerkweg and Jöckel, 2012a). COSMO-CLM (COSMO model in Climate Mode) is the community model of the German regional climate research community jointly further developed by the CLM-Community (Rockel et al., 2008). New boundary conditions (for dynamics, chemistry and contributions) are provided at every time step of the driving model (e.g. EMAC or COSMO-CLM/MESSy) to the finer resolved model instances (COSMO-CLM/MESSy). Accordingly, the MECO(n) model allows for a consistent zooming from the global scale into specific regions of interest.

The simulations analysed in the present study are the same simulations as described in detail by Mertens et al. (2020). Therefore, we present only the most important details of the model set-up. Table 2 lists the used MESSy submodels. The global model EMAC is applied at a resolution of T42L31ECMWF, corresponding to a quadratic Gaussian grid of approx. $2.8° \times 2.8°$ and 31 vertical hybrid pressure levels from the surface up to $10\,\mathrm{hPa}$. The timestep length is set to 720 seconds. To achieve a higher resolution we apply two COSMO-CLM/MESSy nesting steps. The first refinement covers Europe with a horizontal resolution of $0.44°$ and 240 seconds time step length, while the second refinement covers Germany with $0.11°$ horizontal resolution and 120 seconds time step length. Both refinements feature 40 vertical levels from the surface up to

22 km. In the following, the abbreviation CM50 (COSMO(50 km)/MESSy) corresponds to the first refinement (with roughly 50 km resolution) and CM12 (COSMO(12km)/MESSy) corresponds to the second refinement (roughly 12 km resolution). The MESSy submodel MECCA (Sander et al., 2011) is applied in EMAC and COSMO-CLM/MESSy for the calculation of chemical kinetics. The chemical mechanism includes the chemistry of ozone, methane and odd nitrogen. Alkynes and aromatics are not taken into account, but alkenes, and alkanes are considered up to $C_4$. The Mainz Isoprene Mechanism (MIM1, Pöschl et al., 2000) is applied for the chemistry of isoprene and some non-methane hydrocarbons (NMHCs). The complete namelist set-ups and the mechanisms of MECCA and SCAV (scavenging of traces gases by clouds and precipitation, Tost et al., 2006a, 2010) are part of the supplement.

Anthropogenic, biomass burning, agricultural waste burning (AWB) and biogenic emissions are prescribed from external data sources (see Sect. 2.2). Emissions of soil $NO_x$ are calculated on-line (i.e. during model runtime) following the parametrisation of Yienger and Levy (1995). The same applies for emissions of biogenic VOCs which are calculated following Guenther et al. (1995), and emissions for lightning-$NO_x$ for which the parametrisation of Price and Rind (1994) is applied.

The simulation period ranges from 07/2007 to 01/2011. The period in 2007 is the spin-up phase and the years 2008–2010 are analysed. For reasons of computational costs CM12 has been initialised in May 2008 from CM50 and integrated for the period 05/2008-08/2008 only. Therefore, results of CM12 are analysed for JJA 2008 only. To facilitate a one to one comparison with observations EMAC is 'nudged' by Newtonian relaxation of temperature, divergence, vorticity and the logarithm of surface pressure (Jöckel et al., 2006) towards ERA-Interim (Dee et al., 2011) reanalysis data of the years 2007 to 2010. The sea surface temperature and sea ice coverage are prescribed from ERA-Interim as well. CM50 and CM12 are not nudged, but forced at the lateral and top boundaries against the driving model (e.g. EMAC for CM50 and CM50 for CM12).

One feature of chemistry-climate models is the coupling between chemistry, radiation and atmospheric dynamics, meaning that even small changes in the chemical state of the atmosphere lead to changes in the dynamics (which in turn feed back to the chemistry). This feedback can prevent a quantification of the influence of small emission changes on the atmospheric composition. To overcome this issue Deckert et al. (2011) proposed a so called **q**uasi **c**hemistry **t**ransport **m**odel mode (QCTM mode) for EMAC, which can also be applied in MECO(n) (Mertens et al., 2016). To achieve the decoupling between dynamics and chemistry, climatologies are used within EMAC: (a) for all radiatively active substances ($CO_2$ , $CH_4$ , $N_2O$, CFC-11 and CFC-12) for the radiation calculations, (b) nitric acid for the stratospheric heterogeneous chemistry (in the submodel MSBM, Multiphase Stratospheric Box Model, Jöckel et al. (2010)) and (c) for OH, $O^1D$ and Cl for methane oxidation in the stratosphere (submodel CH4). In COSMO-CLM/MESSy only the climatology of nitric acid for the submodel MSBM is required. The applied climatologies are monthly mean values from the *RC1SD-base-10a* simulation described by Jöckel et al. (2016).

## 2.1 Tagging method for source attribution

The source attribution of ozone and ozone precursors is performed using the tagging method described in detail by Grewe et al. (2017), which is based on an accounting system following the relevant reaction pathways and applies the generalised tagging method introduced by Grewe (2013).

For the source attribution the source terms, e.g. emissions, of the considered chemical species are fully decomposed in $N$ unique categories. The definition of the ten categories considered in the current study are listed in Table 3. The tagging method is a diagnostic method, i.e. the atmospheric chemistry calculations are not influenced by the tagging method. To minimise the computational resources (e.g. computing time and memory consumption), the tagging is not performed for the detailed

chemistry from MECCA, but for a simplified family concept. The species of the family concept are listed in Table 4.

The production and loss rates and mixing ratios of the chemical species which are required for the tagging method are obtained from the submodel MECCA. Loss processes like deposition are treated as bulk process, meaning that the changes of the relevant mixing ratios due to dry and wet deposition are memorized and later applied to all tagged species according to their relative contributions.

Due to the full decomposition into $N$ categories, the sum of contributions of all categories for one species equals the total mixing ratio of this species (i.e. the budget is closed):

$$\sum_{\text{tag}=1}^{N} O_3^{\text{tag}} = O_3. \tag{1}$$

To demonstrate the basic concept of the generalised tagging method we consider the production of $O_3$ by the reaction of NO with an organic peroxy radical ($RO_2$) to $NO_2$ and the organic oxy radical (RO):

$$NO + RO_2 \longrightarrow NO_2 + RO. \tag{R1}$$

As demonstrated by Grewe et al. (2017) (see Eq. 13 and 14 therein) the tagging method leads to the following fractional attribution:

$$P_{R1}^{\text{tag}} = \tfrac{1}{2} P_{R1} \left( \frac{NO_y^{\text{tag}}}{NO_y} + \frac{NMHC^{\text{tag}}}{NMHC} \right). \tag{2}$$

Here, all species marked with $^{\text{tag}}$ represent the quantities tagged for one specific category (e.g. land transport emissions);
$P_{R1}$ is the production rate of $O_3$ by reaction R1, $NO_y$ and NMHC represent the mixing ratios of the tagged families of $NO_y$ and NMHC, respectively. The denominator represents the sum of the mixing ratios over all categories of the respective tagged family/species. Accordingly, the tagging scheme takes into account the specific reaction rates from the full chemistry scheme. Further, the fractional apportionment is inherent to the applied tagging method as due to the combinatorial approach, every regarded chemical reaction is decomposed into all possible combinations of reacting tagged species.

Some of the categories listed in Table 3 are not directly associated with emission sectors. These categories are stratosphere, $CH_4$ and $N_2O$. All ozone which is formed by the photolysis of oxygen i.e.

$$O_2 + h\nu \longrightarrow O(^3P) + O(^3P), \tag{R2}$$

is labelled as stratospheric ozone.

The degradation of $N_2O$ is a source for $NO_y$ (and a loss of ozone) by the reaction:

$$N_2O + O^1D \longrightarrow 2NO. \tag{R3}$$

The degradation of $CH_4$ is considered as source of $NMHC^{CH_4}$. This refers to the reaction:

$$CH_4 + OH \longrightarrow CH_3O_2 + H_2O. \tag{R4}$$

As discussed recently in detail by Butler et al. (2018) all tagging methods are based on specific assumptions and have specific limitations. The scheme of Grewe et al. (2017), which we apply in the current study, is based on specific assumptions, which differ from other tagging schemes used in regional and global models. One important difference is the question whether ozone formation is attributed to $NO_x$ or VOC precursors. The schemes which are available in the regional models CMAQ

(called CMAQ-ISM, Kwok et al., 2015) and CAMx (called CAMx OSAT, Dunker et al., 2002) use threshold conditions to check, whether ozone formation is $NO_x$ or VOC limited. Depending on this, the ozone production is attributed to $NO_x$ or VOC precursors only. The scheme of Emmons et al. (2012), applied on the global scale, tags only $NO_x$ and therefore ozone production is only attributed to $NO_x$ precursors. Based on the work of Emmons et al. (2012), Butler et al. (2018) presents a scheme, which attributes ozone formation either to $NO_x$ or VOCs (implying that usually 2 simulations, one with $NO_x$ and one

with VOC tagging, are performed). This scheme was also applied by Lupaşcu and Butler (2019) in a regional model simulation over Europe, using the $NO_x$ tagging scheme only. Compared to discussed schemes the scheme of Grewe et al. (2017) attributes ozone production always to all associated precursors (i.e. NOx, $HO_2$ and VOCs) without any threshold conditions.

If the tagging scheme is used in addition to the perturbation approach (see Table 1) to investigate the influence of mitigation options, the approach of Grewe et al. (2017) leads to the effect that in VOC limited regions a $NO_x$ emission reduction of

an emission sector reduces the contribution of that sector, and increases the contribution of the other sectors. In contrast, a reduction of VOC emissions decreases the contribution of the respective sector only. The latter is similar to the approaches integrated in CMAQ or CAMx, which attribute ozone production in the case of a VOC limit to VOC precursors only. Compared to a $NO_x$ tagging, our approach leads to lower contributions of $NO_x$ sources, since they compete, not only with other $NO_x$ sources, but also with VOC sources.

Because of the family concept, which is necessary to keep the memory consumption and the computational costs low, the tagging method applied in our study can lead to some unphysical artefacts. As an example, Grewe et al. (2017) discuss the production of PAN by NMHCs from $CH_4$ degradation. Further, due to the combinatorial approach for instance also NMHCs from stratospheric origin can occur in small amounts, which is also an unphysical artefact. The main reason for this is the definition of the PAN family, which transfers tags from $NO_y$ to NMHCs. Other tagging schemes have specific issues as well.

As an example, the scheme of Emmons et al. (2012) does not neglect the $O_3$-$NO_x$ null cycle, which leads to an overestimation of local sources compared to long range transport sources (see also Kwok et al., 2015). Overall, the impacts of the underlying

assumptions on the results are difficult to quantify. Therefore, it is important to study effects of different emission sources with different methods (at best in the same model framework), in order to understand better the strengths and weaknesses of the different approaches and their impact on the source attribution results.

Besides these general assumptions of the different methods one specific problem occurs when applying ozone source attribution in regional models; the boundary conditions. Usually, regional studies (e.g. Li et al., 2012; Kwok et al., 2015; Valverde et al., 2016; Pay et al., 2019) just tag ozone from lateral and top boundaries as 'boundary ozone' because no boundary conditions including tagged ozone are available. Recently, Lupaşcu and Butler (2019) used results from a previous global model simulation including a $NO_x$ tagging as boundary conditions for a regional ozone source attribution study with WRF-Chem over Europe. As pointed out by Mertens et al. (2020) our approach has no need for results from previous model runs, as in MECO(n) the tagging is performed in all model instances (i.e. in the global model as well as all regional model instances). Thus, consistent boundary conditions are provided for the regional model instances and source categories for contributions from lateral or model top boundaries are not required. In the present study, the tagging method is configured such that we apply only one global tag for every source category. While this allows to investigate the contributions of all global emissions of a specific emission source to ozone, we are not able to separate contributions from local and long range transport (i.e. we cannot separate contributions from, for example, European and Asian land transport emissions to European ozone levels, but we can quantify the contribution of global land transport emissions to European ozone levels).

In the following, we denote absolute contribution of land transport emissions to ozone as $O_3^{tra}$. Analogously, contributions to the family of $NO_y$ and CO are denoted as $NO_y^{tra}$ and $CO^{tra}$, respectively (cf. abbreviations in Table 3). These absolute contributions correspond to the share of the species total mixing ratio which can be attributed to emissions of land transport. Please note, that the given absolute contributions for ozone are always computed by multiplying the relative contributions to odd oxygen with the ozone mixing ratios. These values are slightly lower than the absolute contributions of odd oxygen. Besides the absolute contributions we investigate relative contributions which give the percentage of the contribution to the total mixing ratio of the specie.

## 2.2 Emission scenarios and numerical experiments

To investigate the influence of the uncertainties of anthropogenic emission inventories on the source attribution results, we perform simulations for two anthropogenic emission inventories. The first emission inventory is the MACCity inventory (Granier et al., 2011), a global inventory with 0.5 x 0.5° horizontal resolution which corresponds to the RCP 8.5 emission scenario for the analysed time frame (called MAC in the following). The second emission inventory is named VEU and considers emissions only for the European area (0.0625 x 0.0625° horizontal resolution). It was composed in the DLR project 'Verkehrsentwicklung und Umwelt'. For this emission inventory the German land transport emissions were estimated bottom up by means of macroscopic traffic simulations. Finally, the land transport emissions are estimated by combining the activity data of the traffic simulations with corresponding emissions factors. For the other European countries, as well as for all other emission sectors, a top down approach was applied. More details about the emission inventory are provided by Hendricks et al. (2017). Further details about the preprocessing of the emissions is given in Appendix A of Mertens (2017).

Two different simulations are performed:

    – *REF*: The MAC emission inventory is applied in EMAC and all regional refinements (e.g. CM50 and CM12);

    – *EVEU*: The MAC emission inventory is applied in EMAC and the VEU emission inventory in the regional refinements.

The VEU emission inventory considers only emissions for the sectors land transport, anthropogenic non-traffic (including
landing and take-off (LTO) of airplanes) and shipping. Table 5 lists the total emissions of $NO_x$, CO, VOC, and the ratio of
$NO_x$ to VOC for these emission sectors. In general, the total emissions of the land transport sector are quite similar, while
the emissions of the sectors anthropogenic non-traffic and shipping are lower in the VEU compared to the MAC emission
inventory. Especially the $NO_x$ and VOC emissions are lower by around 30 % and 50 %, respectively. This leads to different
$NO_x$ to VOC ratios for the total anthropogenic emissions between both emission inventories.
The definition of the emission sectors in VEU is different from the definition in MAC. In the VEU emission inventory
LTO emissions are part of the anthropogenic non-traffic sector, but inflight emissions from aircrafts are not considered in
VEU. Therefore, the MAC aviation emissions are also applied in the *EVEU* simulation. To avoid a double accounting of the
LTO emissions, the aviation emissions in MAC are set to zero in the lowermost level in *EVEU*, leading to a reduction of the
aviation emissions of the MAC emission inventory by 0.05 $Tg\ a^{-1}$ (see Table 5). For the emission sectors agricultural waste
burning (AWB), biomass burning, lightning and biogenic we apply the same emissions in both simulations (see Table. 6). Total
emissions for the global model EMAC, and for CM12 are given in the Supplement (see Section S4).

    Figure 1 displays the geographical distribution of the land transport emissions of $NO_x$, CO, and VOC applied in the *REF*
and *EVEU* simulations and the emission differences between both simulations. Shown are only the emissions of EMAC and
CM50, focusing on Europe. The $NO_x$ land transport emissions for CM12 are depicted in the Supplement (Fig. S7). Further,
more detailed figures showing the geographical distribution in CM50 are part of the Supplement (Fig. S8). The emissions of
CM50 are superimposed onto the emissions applied in EMAC, where the MACCity emissions are applied globally. Despite
comparable total emissions between the MACCity and the VEU emission inventory over Europe, the geographical distributions
differ. Generally, the VEU emission inventory features larger emissions near the hot-spots and lower emissions away from the
hot-spots compared to MAC. Further, MAC features larger $NO_x$ emissions especially the Northern part of the British Islands
and in Finland. Emissions of CO are especially larger around Estonia in MAC compared to VEU. Particularly, over Germany,
the Po Valley and parts of Eastern Europe VEU features more emissions of $NO_x$, CO and VOC (see also totals for CM12 in
Table S4). Besides the difference between the emissions applied in CM50 (and CM12) it is important to note, that for the *REF*
and the *EVEU* simulations the same emissions are applied in EMAC. Therefore, the difference (Fig. 1c) is zero in EMAC.

## 2.3 Model evaluation

A model set-up very similar to the one used for the present study was evaluated with observational data by Mertens et al.
(2016). Generally, the comparison showed a good agreement with observations. The biases are similar to comparable model
systems and exhibit a positive ozone bias and negative biases for $NO_2$ and CO. One important reason for these biases is the
too efficient vertical mixing within the COSMO-CLM model. An evaluation of the ozone mixing ratios simulated by *REF* and

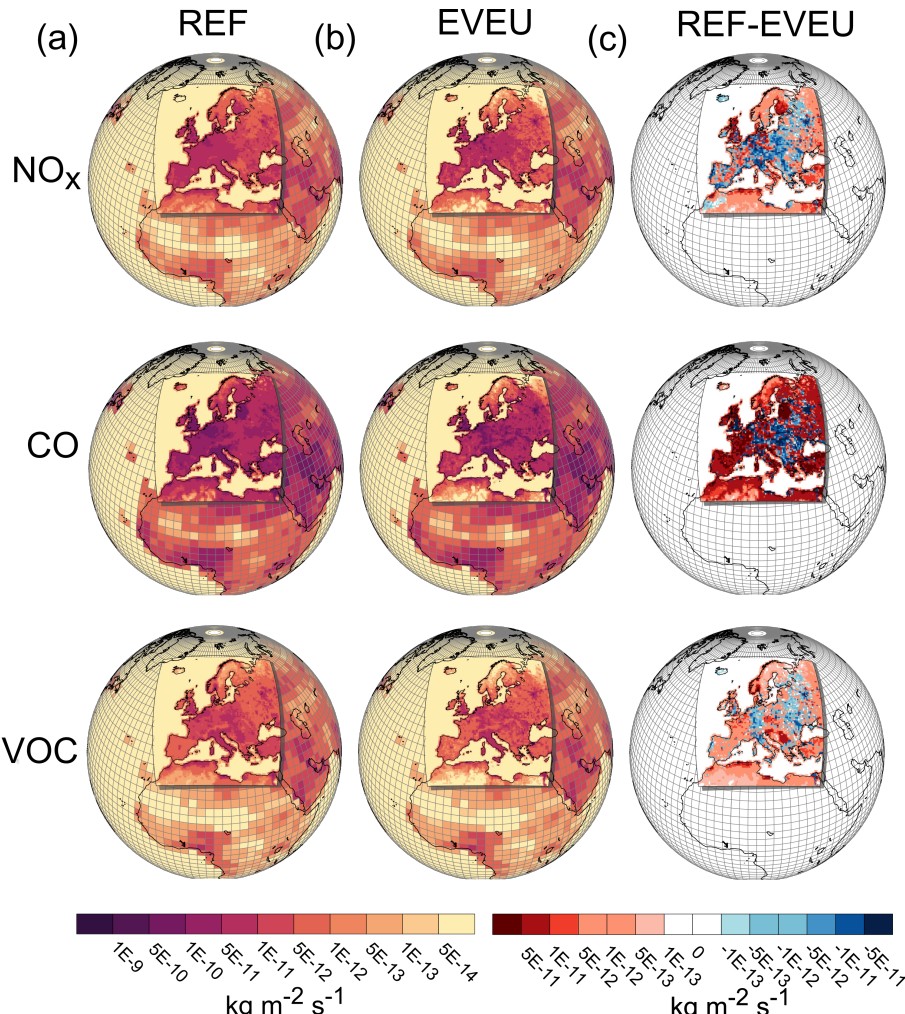

**Figure 1.** Annually averaged emission fluxes (2008 to 2010) from the land transport sector (in $\mathrm{kg\,m^{-2}\,s^{-1}}$). Shown are the emissions as applied in EMAC (based on the MACCity inventory) and in CM50. The emissions of CM50 are superimposed on the emissions of EMAC. In the region covered by CM50 EMAC also uses the MACCity emissions (not visible). (a) the emissions applied in *REF* , (b) the emissions applied in *EVEU* and (c) the difference of the emissions between *REF* and *EVEU* ('REF MINUS EVEU'). Shown are the emission fluxes of $\mathrm{NO_x}$ (in $\mathrm{kg\,NO\,m^{-2}\,s^{-1}}$), CO (in $\mathrm{kg\,CO\,m^{-2}\,s^{-1}}$); and VOC (in $\mathrm{kg\,C\,m^{-2}\,s^{-1}}$).

*EVEU* was presented by Mertens et al. (2020), however, with emphasis on JJA mean values. To investigate the models ability to represent extreme values, we present a brief evaluation of the simulated ozone concentrations in comparison to the Airbase v8 observational dataset (available at, https://www.eea.europa.eu/data-and-maps/data/airbase-the-european-air-quality-database-8, last access 14.2.2020). As the model resolution of $50\,\mathrm{km}$ is too coarse to resolve hot-spots of individual cities we restrict
5    the comparison to those stations which are classified as area types 'suburban' and characterised as 'background'. We focus on

JJA 2008 to 2010 and compare the results to overall 350 measurement stations. The measurements are subsampled at the same temporal resolution (3 hourly) as the model data.

Our comparison with the Airbase v8 data shows the known positive ozone bias (Mertens et al., 2016, 2020). The average root-mean-square-error (RMSE) over all 350 stations is 29.2 µg m$^{-3}$ for *REF*, and 24.3 µg m$^{-3}$ for *EVEU*, respectively. The corresponding mean biases (MBs) are 26.6 % and 20.5 %, respectively (see Table 7). In addition, we calculated also the RMSE and MB for the *REF* simulation considering only measurements and model data at 12 and 15 UTC. For this subsample, both, RMSE and MB decrease considerably. Accordingly, the largest ozone values during daylight are captured very well by the model. As a more detailed comparison between measurements and model result shows, the overestimation of ozone is particularly strong during night. This can partly be attributed to a too unstable boundary layer during night, which is a common difficulty in many models (Travis and Jacob, 2019). In addition, the too strong vertical mixing in the model leads to positive ozone biases at noon and during the night (see also Mertens et al., 2020, 2016). Currently, further investigations are undertaken, about how this bias could be reduced in the future. Besides the too efficient vertical mixing, also too low ozone deposition during night, too low NO or VOC emissions, and successively underestimated ozone depletion during nights could also partly contribute to this bias.

To check the models ability to represent extreme ozone values, the simulated 95th percentiles of ozone is compared with measurements too (see Fig. 2). Overall, the model is able to capture most of the regional variability of the extreme values over Europe. Near the densely populated regions in Benelux, Germany and Italy, however, the model is not able to reproduce the observed 95th percentiles of ozone. In these areas the model resolutions (i.e. also for the 12 km domain, which is not shown here) are too coarse to allow for a representation of extreme ozone values in urban areas. As was shown by prior studies (e.g. Tie et al., 2010) resolutions below 10 km are required to capture high ozone values near cities. Terrenoire et al. (2015) noted that even with 8 km resolution the performance of the applied CHIMERE model is better at rural than at urban sites. This underestimation can also be quantified using the RMSEs and MBs for the 95th percentile which are listed in Table 7.

These results have important implications for the analyses presented in the present study. First of all, the too strong vertical mixing in COSMO-CLM/MESSy leads to a positive bias of the contribution of stratospheric ozone at ground-level. Further, contributions of lightning and aviation at ground-level are likely overestimated due to this overestimated vertical mixing. Altogether COSMO-CLM/MESSy simulates an approximately 1 percentage point lower contribution of anthropogenic emissions to ground-level ozone compared to EMAC (see Mertens et al., 2020).

Due to the coarse model resolution of 50 km our results are representative for the regional scale, but not for specific urban areas. In these urban areas local emissions and local ozone production/destruction might be more important such that contributions of local sources can be much larger than the values we present. On the regional scale, however, Mertens et al. (2020) showed that the results are quite robust w.r.t. the model resolution (down to 11 km).

Because of the stronger ozone bias during night, we further compared the contributions at 12 and 15 UTC with the contributions considering all times of the day. The relative contributions show only small differences, i.e. a slightly larger contribution of anthropogenic emission sources during day (not shown). Therefore, we present always results for all times of the day.

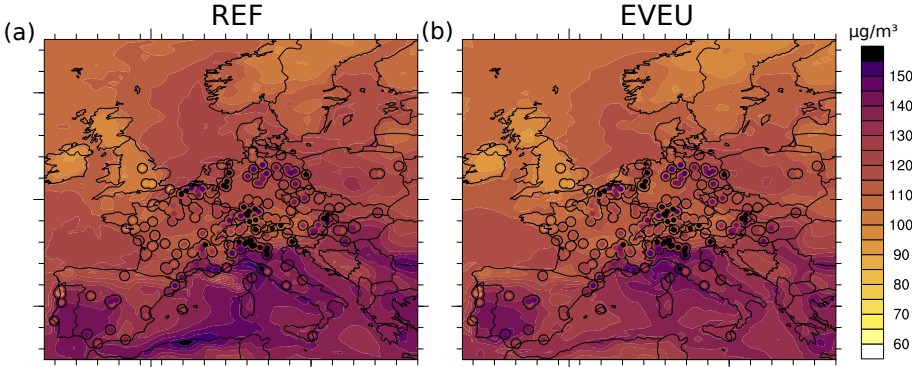

**Figure 2.** 95th percentile of ozone (in μg/m³) for the period JJA 2008 to 2010 as simulated by *REF* (a) and *EVEU* (b). The background colours show the ozone concentrations as simulated by CM50, the circles represent the location of stations of the Airbase v8 observation data. The inner point represents the measured concentrations, the outer point the concentrations in the respective grid box, where the station is located. All values are based on data every 3 hours.

## 3  Contributions of land transport emissions to ground-level mixing ratios of $NO_y$ and CO in Europe

CO and $NO_y$ are direct pollutants of the land transport sector, with different chemical lifetimes. Due to the family concept of the tagging method we investigate contributions to $NO_y$ and not to $NO_x$. Our focus in this section is on the results at the European scale, results of $NO_y$ for Germany will be discussed in Sect. 5. Figure 3 shows $NO_y^{tra}$ for DJF and JJA, respectively. The largest mixing ratios of $NO_y^{tra}$ are simulated near Southern England, the Paris metropolitan region, Western Germany and the Benelux states as well as the Po Valley and the Moscow metropolitan region. In these regions contributions of up to 10 nmol mol$^{-1}$ are simulated. In general, larger absolute contributions occur during DJF compared to JJA, but the seasonal cycle of the land transport emissions is small in both emission inventories (see supplement Figure S4). Accordingly, the differences of $NO_y^{tra}$ between DJF and JJA are likely not caused by seasonal differences of the emissions, but by larger mixing layer heights and a more effective photochemistry during JJA compared to DJF.

The seasonal change of $NO_y^{tra}$ is smaller than differences between *REF* and *EVEU*. Near areas with large land transport emissions *EVEU* simulates 3 to 4 nmol mol$^{-1}$ larger contributions than *REF*. In most of the hot-spot regions (e.g. Paris and the Po Valley) the differences are even larger and the contributions calculated by *EVEU* are 5 nmol mol$^{-1}$ larger than in *REF*. In some regions the results of both simulations are in total contrast. In *REF* for example, absolute contributions of up to 4 nmol mol$^{-1}$ are simulated in Finland, while *EVEU* simulates absolute contributions below 1 nmol mol$^{-1}$.

The relative contribution of land transport emissions to ground-level $NO_y$ is in the range of 40 % to 70 % in most parts of Europe (see Fig. 4). These relative contributions are similar to the share of land transport $NO_x$ emissions to all $NO_x$ emissions (see Fig. S9 in the Supplement), but compared to the share of the emissions the contributions to $NO_y$ are slightly lower near hot-spots, and larger in rural areas.

During DJF, *REF* simulates the lowest relative contributions of 30 to 50 % over most parts of Europe. During summer the contributions increase up to 60 % with the largest values in Southern Germany, the Po Valley, and Southern England. *EVEU* simulates a smaller difference of the contributions between DJF and JJA as *REF*. Further, the maxima are generally slightly larger and contributions of up to 70 % are simulated around the Po Valley and the Paris area. Interestingly, the relative contributions are lower during DJF than during JJA while the absolute contributions are larger during DJF than during JJA. Most likely this is caused by the lower amount of anthropogenic non-traffic $NO_x$ emissions during JJA compared to DJF (see Fig. S4 in the Supplement).

The simulated mixing ratios of $CO^{tra}$ (see Fig. 5) show a similar behaviour as $NO_y^{tra}$, implying that contributions in DJF are larger than in JJA. This seasonal difference is most likely caused by lower mixing layer heights and increased lifetime of CO during DJF compared to JJA, as OH concentrations are lower in winter compared to summer. Generally, the largest contributions are simulated in southern England, around Paris, Western Germany, the Po Valley and around Moscow. In *EVEU* contributions of up to 75 nmol mol$^{-1}$ are simulated around London, Paris, Milan and Moscow, while the results of the *REF* simulation show lower contributions in the Western European regions of mostly 50 to 60 nmol mol$^{-1}$. Compared to $NO_y^{tra}$, however, some hot-spots stand out in the results of the two simulations. *EVEU*, for example, shows larger contributions (40 to 60 nmol mol$^{-1}$) to CO over Hungary or southern Poland. In difference to this, *REF* shows contributions of

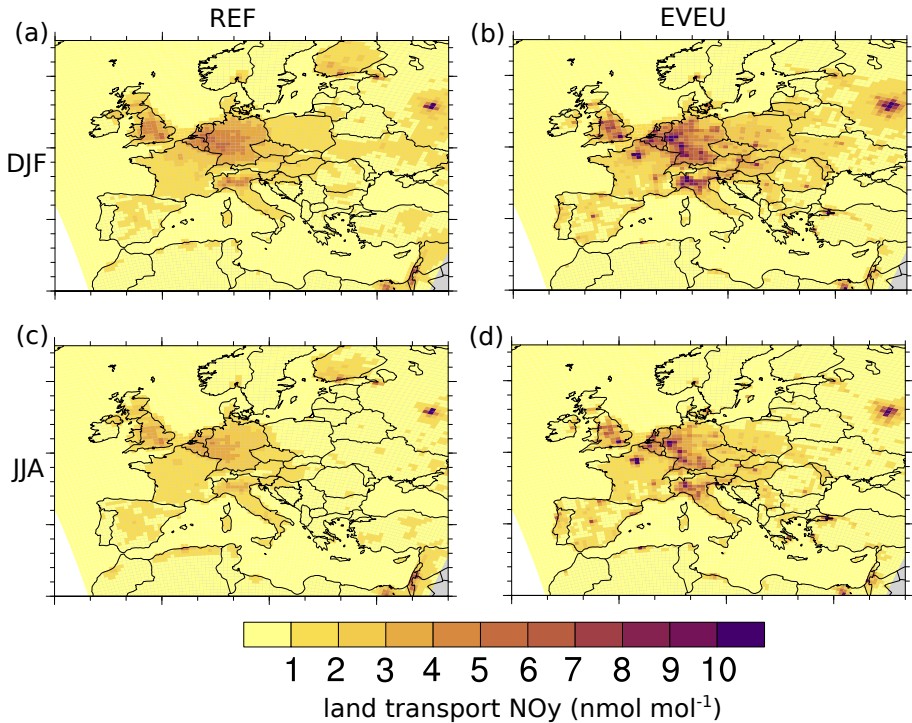

**Figure 3.** Absolute contribution of land transport emissions to ground-level $NO_y$ (in $\mathrm{nmol\,mol^{-1}}$) as simulated by CM50. (a) and (b) contributions for the period DJF (2008 to 2010) of the *REF* and *EVEU* simulations, respectively. (c) and (d) contributions for the period JJA (2008 to 2010) of the *REF* and *EVEU* simulations, respectively.

30 to 50 $\mathrm{nmol\,mol^{-1}}$ over Estonia. These differences between contributions are directly attributable to the differences of the emission inventories (Fig. 1). Hence, the uncertainties with respect to the CO emissions of land transport in these regions are quite large.

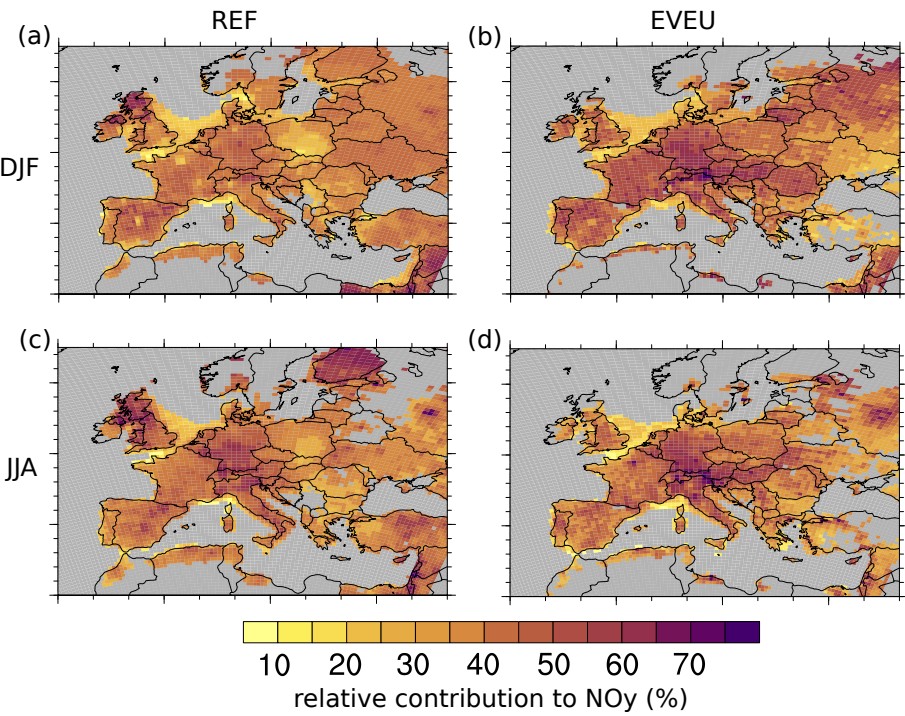

**Figure 4.** Relative contribution of land transport emissions to ground-level $NO_y$ (in %) as simulated by CM50. (a) and (b) contributions for the period DJF of the *REF* and *EVEU* simulations, respectively. (c) and (d) contributions for the period JJA of the *REF* and *EVEU* simulations, respectively. Grey areas indicate regions where the absolute $NO_y$ mixing ratios are below 0.5 nmol mol$^{-1}$. In these regions no relative contributions are calculated for numerical reasons.

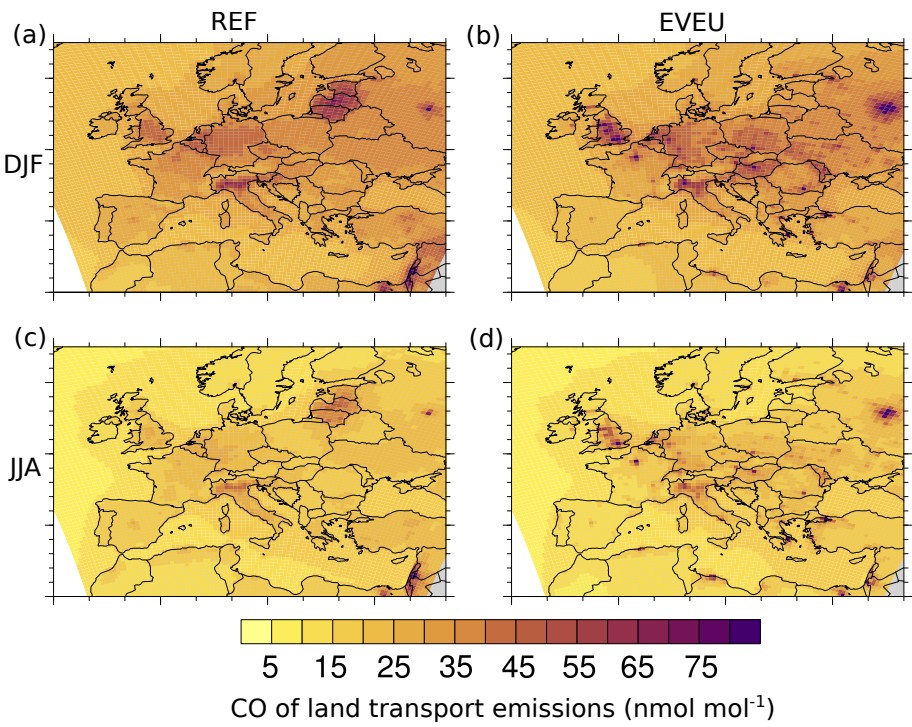

**Figure 5.** Absolute contribution of land transport emissions to ground-level CO (in $\mathrm{nmol\ mol^{-1}}$) as simulated by CM50. (a) and (b) contributions for the period DJF of the *REF* and *EVEU* simulations, respectively. (c) and (d) contributions for the period JJA of the *REF* and *EVEU* simulations, respectively.

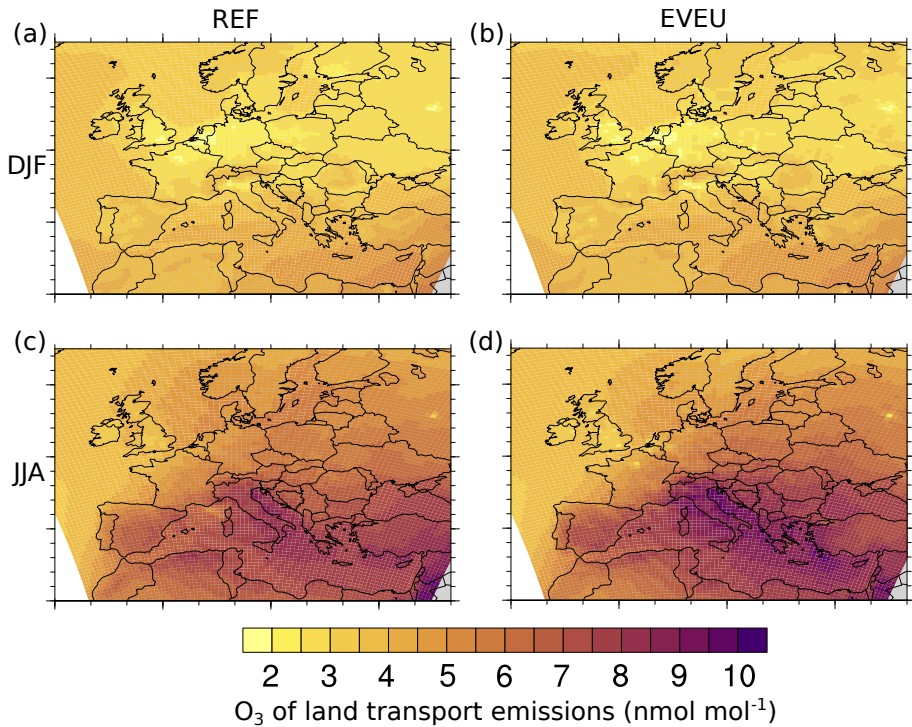

**Figure 6.** Absolute contribution of land transport emissions to ground-level $O_3$ (in $nmol\ mol^{-1}$) as simulated by CM50. (a) and (b) contributions for the period DJF of the *REF* and *EVEU* simulations, respectively. (c) and (d) contributions for the period JJA of the *REF* and *EVEU* simulations, respectively.

## 4 Contribution of land transport emissions to ozone in Europe and Germany

In contrast to $NO_y$ and CO, ozone is a secondary pollutant and emissions have an indirect effect on it. Therefore, this section quantifies the contribution of land transport emissions to ozone in detail. Besides land transport emissions, however, many other sources contribute to ground-level ozone. Generally, the most important sources which contribute globally to ozone are

downward transport from the stratosphere, anthropogenic non-traffic, shipping, lightning and biogenic emissions (e.g. Lelieveld and Dentener, 2000; Grewe, 2004; Hoor et al., 2009; Dahlmann et al., 2011; Emmons et al., 2012; Grewe et al., 2017; Butler et al., 2018). Table 8 lists the contributions of different emission sources to ozone for Europe averaged for JJA 2008 to 2010 and for the results of *EVEU* and *REF* (see also Fig. S6 for zonally averaged vertical profiles of the contributions). The most important sources for ground-level ozone in Europe are biogenic emissions ($\approx 19$ %), anthropogenic non-traffic emissions

($\approx 16$ %), methane degradation ($\approx 14$ %) and land transport emissions ($\approx 12$ %). With increasing height the contributions of ground based emission sources decrease, accordingly the contribution of land transport emissions decrease to $\approx 8$ % at 600 hPa. At the same time the importance of ozone transported downward from the stratosphere, lightning and aviation increases. At a height of 200 hPa more than 50 % of the ozone is from stratospheric origin. The contribution of land transport emissions drops to around 3 %. Further, the differences between the results of *REF* and *EVEU* decrease with increasing height,

indicating the larger importance of long range transport. The latter is equal in both simulations due to identical emissions for the global models and therefore identical boundary conditions for CM50.

### 4.1 Seasonal average contribution to ground-level ozone

DJF ground-level $O_3^{tra}$ simulated by *REF* and *EVEU* (see Fig. 6) ranges between 2 to 4 nmol mol$^{-1}$. Even lower ground-level $O_3^{tra}$ is simulated near some hot-spots due to ozone titration. The absolute contributions mentioned above correspond to relative

contributions of $O_3^{tra}$ of around 8 % over large parts of Europe (see Fig. 7). Although the European emission inventories differ in the simulations, the contributions (absolute and relative) show almost no differences. The emissions of the global model, however, are identical in *REF* and *EVEU* leading to identical contributions at the boundaries of the regional domain. Hence, the contributions during DJF are mainly dominated by long range transport towards Europe which was reported by Karamchandani et al. (2017). This is caused by the low ozone production and long lifetime of ozone during winter.

During JJA the ozone production increases and local emissions play a larger role. Therefore, $O_3^{tra}$ increases to 5 to 10 nmol mol$^{-1}$, implying an increase of the contributions to 10 to 16 %. The geographical distribution of the contribution is similar for both emission inventories, showing increasing absolute and relative contributions from North-West to South-East. The largest relative contributions are simulated around the Po Valley while the largest absolute contributions are shifted downwind from Italy to the Adriatic Sea. In these regions the differences between the results of the two simulations are largest, reaching up

to 2 nmol mol$^{-1}$ for the absolute and 2 percentage points for the relative contributions, respectively. The larger differences between the results of *REF* and *EVEU* during summer compared to winter are mainly caused by the increasing ozone production over Europe during spring and summer. Accordingly, differences in the emission inventories modify the regional ozone budgets more efficiently.

To quantify the contributions of land transport emissions and other emission sources in different regions in more detail, Fig. 8 shows area-averaged relative contributions for JJA and DJF for the *REF* and *EVEU* simulations (absolute contributions are given in Table S1 to Table S8 in the Supplement). The geographical regions were defined according to the definitions of the PRUDENCE project (Christensen et al., 2007). However, we performed some slight modifications. The region Alps

was split up in two separate regions called 'Northern Alps', defined as rectangular box ($46° : 48°$ N and $9° : 13°$ E), and 'Po Valley' ($44° : 46°$ N and $5 : 15°$ E). Note, however, that the region Northern Alps contains parts of Switzerland and Southern Germany, which are still rather flat and subject to large land transport emissions. In addition, we defined a region called 'inflow' ( $40° : 60°$ N and $-13° : -11°$ E). This region is used to quantify contributions in the air advected towards Europe. A figure summarizing the definition of all regions is part of the Supplement (Fig. S12).

The relative contribution of $O_3^{tra}$ in the 'inflow' region is about 9 % in both seasons and for both European emission inventories. During DJF the contributions in all regions are very similar. During JJA the contribution of land transport emissions increases in most regions compared to the 'inflow' ($\approx 9$ %). In the Po Valley $O_3^{tra}$ reaches up to 16 %. Unfortunately, the difference between $O_3^{tra}$ in a specific region and $O_3^{tra}$ in the corresponding region 'inflow' cannot be used to calculate $O_3^{tra}$ from European emissions. Such a calculation requires different tags for global and European land transport emissions. The

relative contribution of other anthropogenic emissions (anthropogenic non-traffic, shipping, and aviation, see also Table 3) in the 'inflow' region ($\approx 34$ %) is also very similar in both seasons. During DJF the contributions over all regions in Europe are very similar to the contribution in the 'inflow' region. During JJA, in contrast, a West-East gradient of the anthropogenic contributions is present over Europe with a decrease down to $\approx 27$ % in Eastern Europe. This decrease is mainly due to the seasonality of the different emissions (discussed further below). The biogenic emissions category shows different relative

contributions in the 'inflow' region during DJF ($\approx 11$ %) compared to JJA ($\approx 14$ %). This is mainly caused by the strong increase of biogenic emissions during summer compared to winter. In the different regions the relative contributions increase during JJA compared to DJF, and, compared to the 'inflow' up to $\approx 20$ %. The contribution of all other tagging categories during DJF is around $\approx 47$ % in most regions, and ranges between 41 % and 36 % during JJA.

As already discussed, the emissions of the land transport sector show almost no seasonal cycle (Fig. S4 in the Supplement),

while the absolute and relative contribution of $O_3^{tra}$ shows a seasonal cycle. This seasonal cycle is caused by a complex interplay of the seasonal cycles of different emission sources, meteorology and photochemical activity. The seasonal cycle of the relative contribution of $O_3^{tra}$ is shown in Fig. 9. The seasonal cycle of the absolute contribution is similar to the cycle of the relative contribution, but shows the largest peak during June where the absolute ozone levels are largest (see Fig. S10 in the Supplement). The contribution peaks between May to July and in October ($\approx 13$ % averaged over Europe for the column

up to 850 hPa) and has a minimum of 9 % during December to March. The decrease of the contribution during the summer months is mainly caused by the large contribution of biogenic emissions (biogenic VOCs and soil-$NO_x$) during July and August and subsequent increasing contributions of $O_3^{soi}$. The decrease of the contribution during DJF is mainly caused by increasing contributions from the stratosphere and anthropogenic non-traffic emissions. The categories show a strong seasonal cycle with peaks of the contributions during March and May (Fig. S3 in the Supplement). The indicated standard deviation

of the contribution shows that in winter, spring, and autumn the year to year variability (blue shading) is the most important

source of uncertainty. Here, differences in regional emissions lead only to small differences (orange shading). During summer, however, the differences of the regional emissions strongly contribute to the uncertainties.

The differences between the extreme absolute and relative contributions of $O_3^{\mathrm{tra}}$ between *REF* and *EVEU* (expressed as 95th percentile) are larger than for the mean values. The 95th percentile of the relative contribution of $O_3^{\mathrm{tra}}$ to ground-level ozone reaches up to 24 % in the Po Valley using the VEU emission inventory (see Fig. 10). In *REF* the maxima are lower by 4 to 5 percentage points compared to *EVEU*. In contrast to the mean values, the extreme values occur mainly near the regions with the largest land transport emissions, namely over France, Italy and Germany. Over France and Germany extreme values (depending on the applied emission inventory) in the range of 16 to 18 % occur, while the values in Northern Italy range from 20 to 24 %.

Focussing on Germany, the relative contribution of $O_3^{\mathrm{tra}}$ to ground-level ozone is 10 to 15 %. The contribution has a North-West to South-East gradient. One important contributor to this gradient are the strong shipping emissions in the English Channel, North- and Baltic- sea (e.g. Matthias et al., 2010). These emissions lead to larger relative and absolute contributions of shipping emissions in Northern and Western Germany, which decrease towards the South. The absolute contributions are around 2 to 3 $\mathrm{nmol\,mol^{-1}}$ during DJF and 4 to 6 $\mathrm{nmol\,mol^{-1}}$ during JJA (averaged for 2008 to 2010). The largest 95th percentile of the relative contribution of land transport emissions is simulated in Southern Germany (up to 22 %).

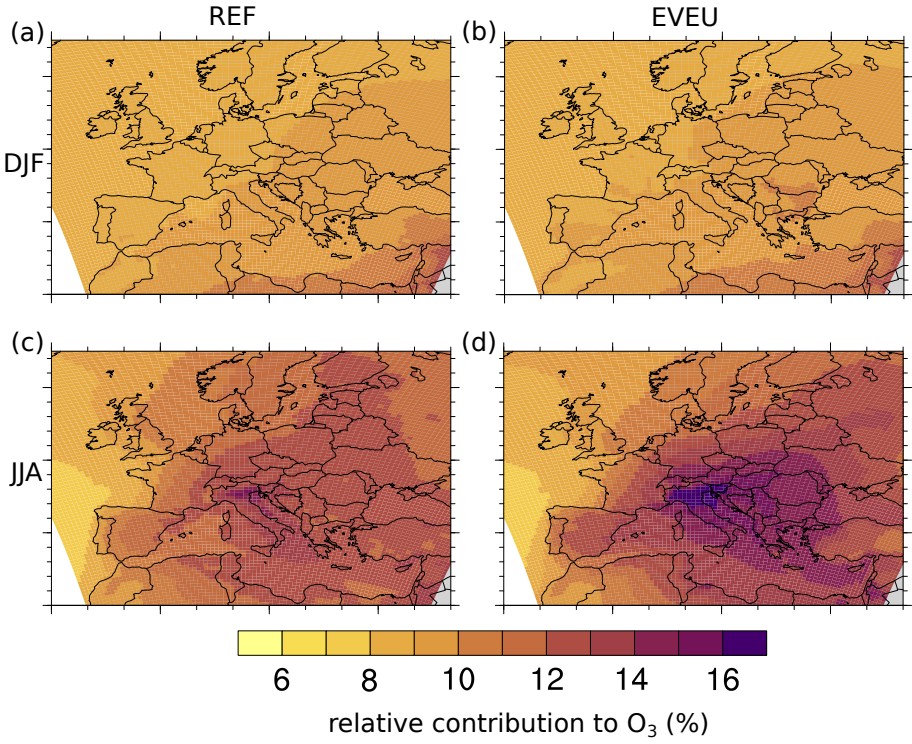

**Figure 7.** Relative contribution of land transport emissions to ground-level $O_3$ (in %) as simulated by CM50. (a) and (b) contributions for the period DJF of the *REF* and *EVEU* simulations, respectively. (c) and (d) contributions for the period JJA of the *REF* and *EVEU* simulations, respectively.

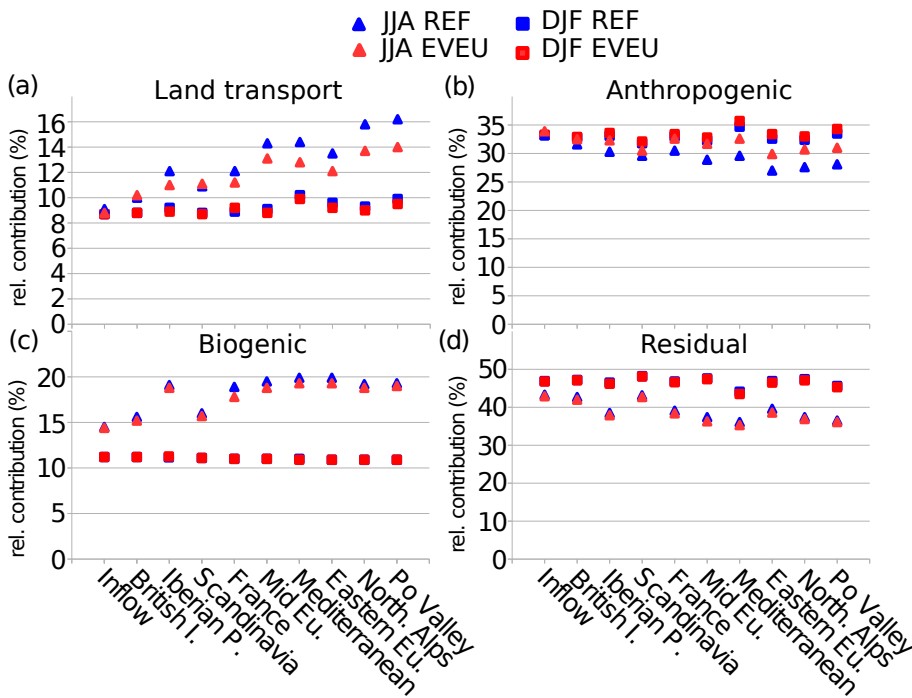

**Figure 8.** Relative contributions to ground-level ozone (in percent) area averaged in different geographical regions for DJF 2008 to 2010 (triangles) and JJA 2008 to 2010 (squares). Shown are the results of the *REF* (blue) and the *EVEU* simulations (red) for (a) the land transport category, (b) the anthropogenic emissions, (c) the biogenic category, and (d) all other categories. For simplicity the anthropogenic contains the categories anth. non-traffic, aviation and shipping. The residual contains all other categories. The vertical-axis scale differs for (a) to (d).

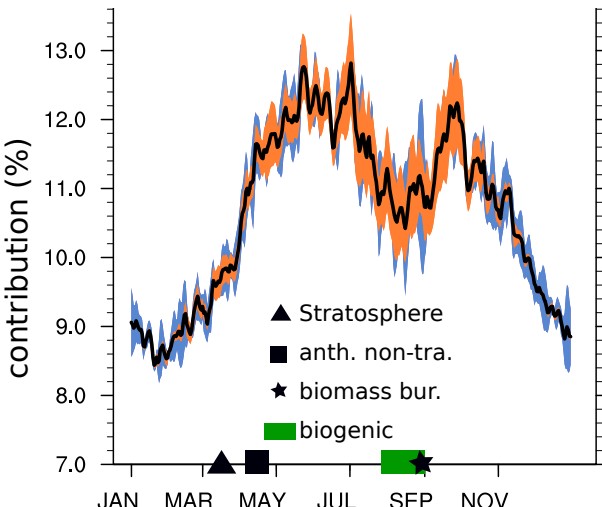

**Figure 9.** Seasonal cycle of the relative contribution of $O_3^{tra}$ to the ozone column up to 850 hPa (in %). The black line indicates the mean contribution as simulated by CM50, averaged over the years 2008–2010 and the two simulations (*REF*, *EVEU*). The blue shading indicates the standard deviation with respect to time for the years 2008 to 2010 for the *EVEU* simulation. The orange shading indicates the standard deviation with respect to time between the 2008–2010 averaged seasonal cycles of the *REF* and the *EVEU* simulations. The symbols on the horizontal axis indicate the time frames during which the categories named stratosphere, anthropogenic non-traffic (anth. non-tra.), biomass burning (biomass bur.) and biogenic contribute the most.

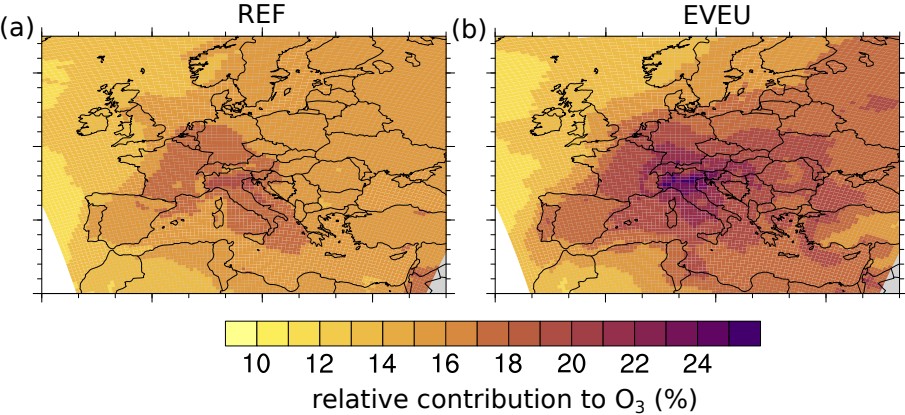

**Figure 10.** 95th percentile of the relative contribution of $O_3^{tra}$ (in %) as simulated by CM50 based on 3-hourly model output. (a) and (b) contributions for JJA of the *REF* and *EVEU* simulations, respectively.

## 4.2 Contribution during extreme ozone events

To better characterise episodes of extreme ozone values, it is important to know which emission sources contribute to and/or drive these extreme ozone values. Therefore, we investigate the contribution of land transport emissions during extreme ozone episodes. As discussed in Sect. 2.3 the contributions we report are representative on the regional scale. For analyses of the local scale (i.e. individual cities) the resolution of the model is too coarse.

First, the 99th, 95th and 75th percentiles of the ozone mixing ratios for the period JJA 2008 to 2010 are calculated (based on 3-hourly model output, see Figs. S1 and S2 in the Supplement). Second, the categories contributing to these 99th, 95th and 75th percentiles of ozone are analysed. Generally, the contributions to these extreme values have a high spatial variability. To capture this spatial variability, the contributions are analysed for the whole CM50 domain as well as for specific regional subdomains as introduced in Sect 4.1.

The range of contributions in the different regions is shown in Fig. 11. Generally, the relative contribution of $O_3^{tra}$ (Fig. 11a and b) increases for increasing ozone percentiles in most regions. This increase is largest in the regions Po Valley, Northern Alps, Mid Europe, France and the British Islands. The largest contributions of $O_3^{tra}$ occur in the Mediterranean region, Northern Alps, Po Valley, Mid Europe and France. Especially in these regions, *EVEU* simulates larger median and maximum relative contributions of $O_3^{tra}$ compared to *REF*. Further, the range of contributions for almost all regions is larger in *EVEU* compared to *REF*. The ozone values at the 95th percentile (see Sect. 2.3) and at the other percentiles (see Figs. S1 and S2 in the Supplement), however, are similar for *REF* and *EVEU* (i.e. none of the emission inventories leads to strongly different representations of extreme ozone events in the model). Accordingly, the discussed differences of the relative contributions are not caused by a different representation of the ozone values themselves, but only due to the different geographical and sectoral distributions of the emissions in *REF* and *EVEU*. This demonstrates the large uncertainty, especially for contributions during high ozone events, of the source attribution analyses which is caused by the uncertainties of emissions inventories (e.g. geographical distribution of emissions, total emissions per sector). These uncertainties must be taken into account in source attribution studies focusing on high ozone events.

For the 99th percentile of ground-level ozone the median of the relative contributions of $O_3^{tra}$ in the region Po Valley is around 17 % / 22 % (*REF/EVEU* simulation), while the 95th percentile is around 18 % / 25 %. The contributions in the region Northern Alps are only slightly smaller, as parts of Southern Germany and Switzerland with large land transport emissions are also part of this region. The region with the third largest contributions is Mid Europe (including mainly Germany and the Benelux States). Here, median contributions (at 99th percentile of ozone) of 16 % / 18 % and contributions (at 95th percentile) of 18 % / 23 % are simulated. The largest contributions (between 24 and 28 % for the *EVEU* simulation) are mainly simulated in the Po Valley, in South-Western Germany, Western Germany and around Paris. For the lower percentiles of ground-level ozone the contribution of land transport emissions decreases and reach median contributions of 13 to 16 % and 95th percentiles of 15 to 21 % in the regions Mediterranean, Alps, Mid Europe and France.

The medians of the relative contribution of other anthropogenic emissions (i.e. the emission sectors anthropogenic non-traffic and aviation) range in all regions from 17 to 25 % (Fig. 11c and d). Hence, the contribution of other anthropogenic emissions is

larger than the contribution of land transport emissions. The increase of the contribution of other anthropogenic emissions with increasing ozone percentiles, however, is lower compared to the increase of $O_3^{\mathrm{tra}}$. Accordingly, the relative importance of land transport emissions increases with increasing ozone values and hence land transport emissions are an important driver of large ozone values. This is in general in line with Valverde et al. (2016) who found that concentration peaks of ozone in Barcelona and Madrid can be explained by ozone attributed to road transport emissions. However, their contributions are in general much larger than the contributions we found (see more details in Sect. 7). Besides the contribution of land transport emissions, however, also the relative contribution of biogenic emissions to ozone increases with increasing ozone levels (Fig. 11e and f). Therefore, also biogenic emissions play an important role during high ozone values.

While the relative contributions to ozone of the shown categories increase with increasing ozone levels, the contribution of the shipping emissions and all other categories decrease with increasing ozone levels in almost all regions (Fig. S5 in the Supplement). Only in the Mediterranean region *REF* simulates also an small increase of the relative contribution of shipping emissions with increasing ozone levels.

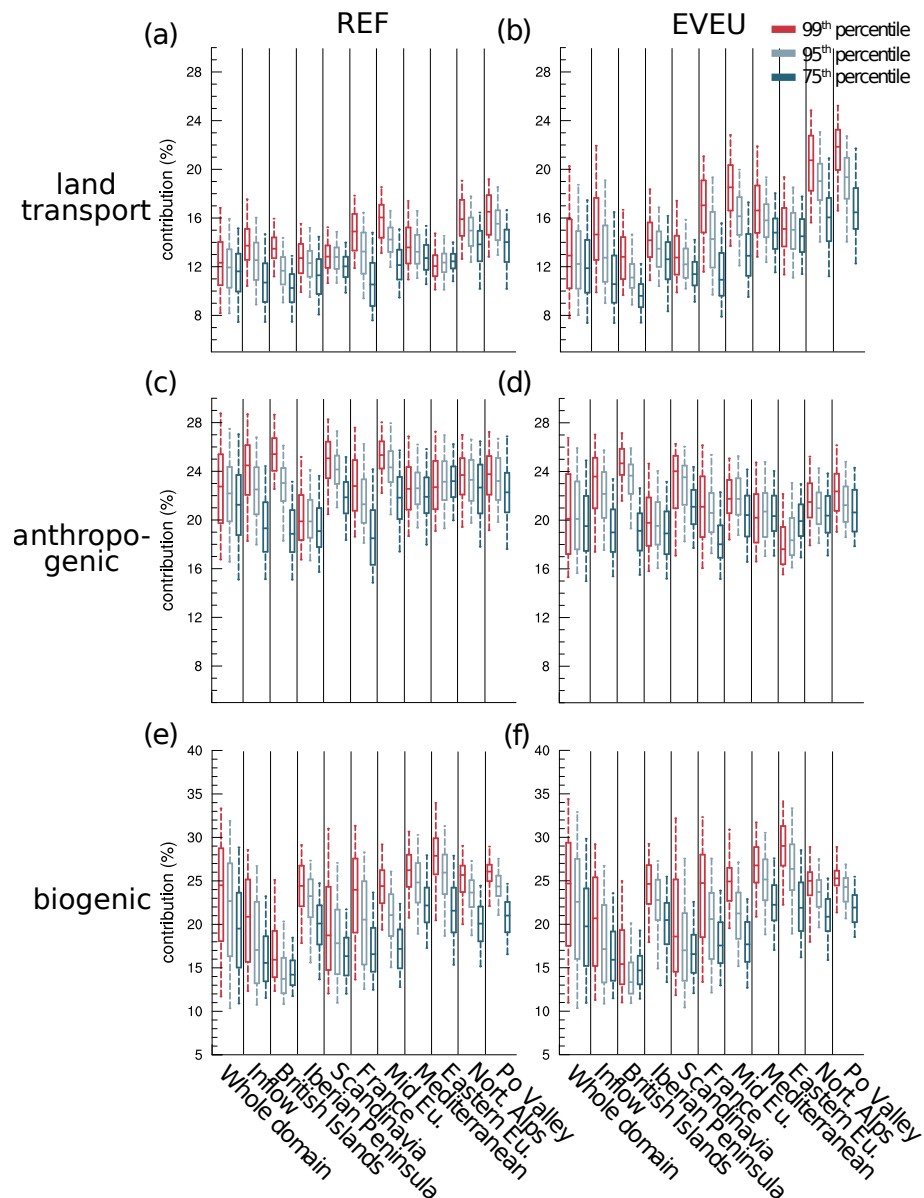

**Figure 11.** Box-whisker plot showing the contributions of the most important emission sources at the 99th, 95th and 75th percentile of ozone as simulated by CM50. For simplicity only the contributions for land transport, biogenic and other anthropogenic emissions (anthropogenic non-traffic, and aviation) to ground-level ozone (in %) are shown. Therefore, the contributions do not add up to 100 %. (a) and (b) show the relative contribution of $O_3^{tra}$; (c) and (d) the relative contribution of anthropogenic emissions (anthropogenic non-traffic and aviation); and (e) and (f) the relative contribution of $O_3^{soi}$. The lower and upper end of the box indicate the 25th and 75th percentile, the bar the median, and the whiskers the 5th and 95th percentile of the contributions of all gridboxes within the indicated region. All values are calculated for JJA of the period 2008 to 2010 and are based on 3-hourly model output. The data are transformed on a regular grid with a resolution of $0.5°$ x $0.5°$ to allow for the regional analyses.

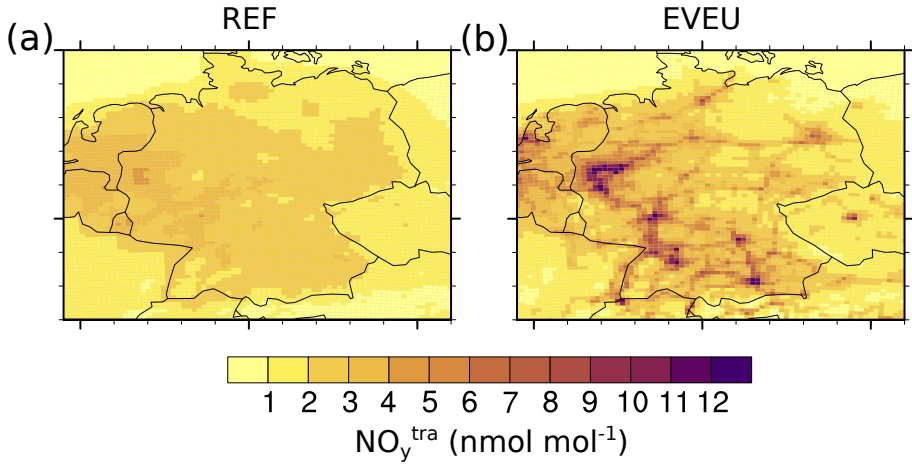

**Figure 12.** Absolute contribution of $NOy^{tra}$ (in nmol mol$^{-1}$) for JJA 2008 of land transport emissions as simulated by CM12. (a) and (b) show contributions for the period JJA of the *REF* and *EVEU* simulations, respectively.

## 5 Contribution of land transport emissions to reactive nitrogen in Germany

So far the results of the European domain are analysed. The resolution of the VEU emission inventory, however, is much finer (roughly 7 km) as the resolution of the European domain. Accordingly, the full potential of the emission inventory is not revealed. Therefore, this section is dedicated to the results of CM12 focusing on Germany. As shown by Mertens et al. (2020) the contribution of land transport emissions to ozone in Germany changes only slightly, when the model resolution is increased from 50 km to 12 km. The changes due to the increase in the resolution are smaller than the differences between the results of both emission inventories. Therefore, we focus on the contribution of land transport emissions to $NO_y$ where the results depend stronger on the model resolution. The results of $O_3^{tra}$ for Germany is discussed at the end of Sect. 4.1.

Figure 12 shows the absolute contribution of $NO_y^{tra}$ for JJA 2008 as simulated by CM12. As already discussed, the differences between the two emission inventories are rather large. The *REF* simulation shows maximum contributions of around 5 nmol mol$^{-1}$, while the *EVEU* simulation shows contributions of up to 12 nmol mol$^{-1}$. These large values occur around the large cities in Bavaria (Munich, Nuremberg) and the large cities in (South-)Western Germany (Stuttgart, Frankfurt, Rhine-Ruhr area). These results indicate the importance of land transport emissions for the mixing ratios of reactive nitrogen levels in German cities. Further, they clearly show the importance of fine resolved emission inventories (and models) for source attribution of short lived chemical species.

## 6 Contribution of land transport emissions to net ozone production in Europe

We analyse the contribution of land transport emissions to the ozone budget in Europe by investigating the net ozone production, which is defined as:

$$P_{O3} = ProdO3 - LossO3, \tag{3}$$

with production ($ProdO3$) and loss rates ($LossO3$) of ozone as diagnosed by the tagging method for the different tagged categories (see Supplement Sect. S5).

According to the analysis (see Table 9) the land transport emissions are the second most important anthropogenic emission sector contributing to $P_{O3}$ in Europe. In general, the results obtained with both emission inventories are rather similar, caused by similar emissions of the land transport sector. For both simulations $P_{O3}$ due to land transport emissions integrated over the CM50 domain and up to 850 hPa is around 13 $Tg(O_3) \, a^{-1}$. $P_{O3}$ integrated up to 200 hPa is 23 $Tg(O_3) \, a^{-1}$.

The differences between contributions of $O_3^{tra}$ discussed in Sect. 4 are mainly caused by the differences of the total emissions of the anthropogenic non-traffic sector. $P_{O3}$ of the anthropogenic non-traffic category differs by roughly 30 % between REF and EVEU, whereas the total net ozone production differs by roughly 15 %. Due to the lower total emissions in the VEU emission inventory compared to the MAC inventory, less ozone is produced in the former.

The regions where ozone is predominantly formed by land transport emissions are displayed in Fig. 13a and Fig. 13b, showing the relative contribution of land transport emissions to $P_{O3}$. Here, the analysis is restricted to the period May to September where $P_{O3}$ is largest. Additionally, Fig. 13c and Fig. 13d indicate the emission sectors which contribute most to $P_{O3}$ up to 850 hPa in the respective gridbox. Consistent with previous analyses the results show that the relative contribution of land transport emissions to $P_{O3}$ is in general larger in *EVEU* compared to *REF*. The contribution is lowest over the Atlantic and along the main shipping routes in the Mediterranean Sea. In these regions ozone up to 850 hPa is mainly formed from shipping emissions (Fig. 13c and d). Generally, the contribution of land transport emissions to $P_{O3}$ is largest over Central Europe, including parts of the Iberian Peninsula, the British Islands and Italy. In these regions the contributions range from 25 % to 35 % in *REF* and 25 % to 40 % in *EVEU*. Further, the regions of large contributions extend much more to the East (including Austria and Hungry) in *EVEU* compared to *REF*. Besides these regions the contributions of land transport emissions to $P_{O3}$ range from 15 % to 20 % in most areas. However, both simulation results indicate regions especially in Northern Europe, but also in the Mediterranean Sea and Africa with very large contributions (above 35 %). These regions, however, generally show low absolute values of $P_{O3}$. Therefore, the large contribution of land transport emissions is not very meaningful.

With contributions from 25 % to 40 %, land transport emissions contribute significantly to the ozone production up to 850 hPa. However, in only very few regions (Western Germany, Austria and Northern Italy) and only in *EVEU* land transport emissions are the most important contributor to $P_{O3}$ (Fig. 13).

Outside these regions the results of *REF* and *EVEU* show that biogenic emissions are most important over the Iberian Peninsula, large parts of Eastern Europe, and Africa. For Central Europe and Northern Europe the *REF* results indicate that the anthropogenic non-traffic category is most important, while the *EVEU* results indicate biogenic and land transport as the most

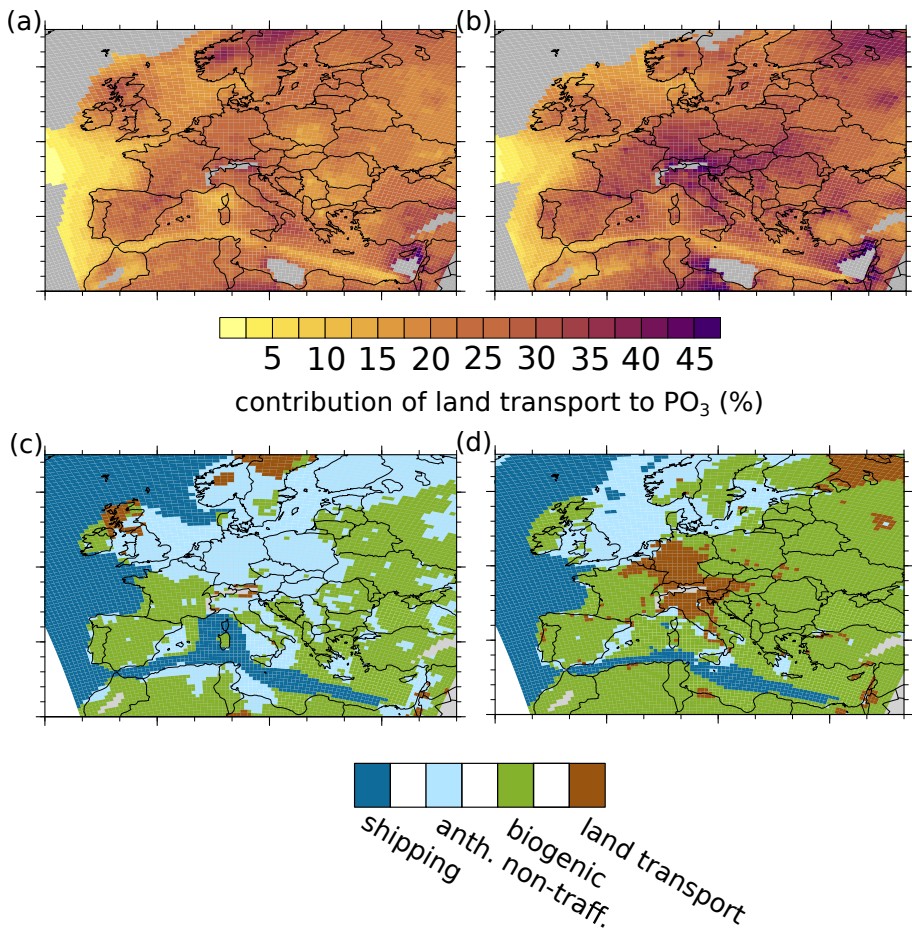

**Figure 13.** Contribution analysis for $P_{O3}$ integrated from the surface up to 850 hPa. (a) and (b) show the relative contribution of land transport emissions to $P_{O3}$ (in %) for the *REF* and the *EVEU* simulation, respectively. (c) and (d) indicate the emission sectors which contribute most to $P_{O3}$ up to 850 hPa, for the *REF* and *EVEU* simulation, respectively. Analysed are averaged data for the period May–September 2008 to 2010 as simulated by CM50. Grey areas in (a) and (b) indicate regions where $P_{O3}$ is below $1.5 \cdot 10^{-13}$ mol mol$^{-1}$ s$^{-1}$. In these regions no relative contributions are calculated for numerical reasons.

important. This underlines that the uncertainty of such analysis is strongly influenced by the uncertainties of the anthropogenic and biogenic emission inventories (or parametrizations to calculate these emissions).

## 7 Discussion

Our analyses demonstrate the importance of land transport emissions to European reactive nitrogen ($NO_y$) mixing ratios. The largest contribution of land transport emissions to $NO_y$ are simulated in Southern England, Benelux, Rhine-Ruhr, Paris and the Po Valley. These regions correspond well with the regions where ground-level measurements, satellite observations or air-
quality simulations report the largest nitrogen dioxide levels (e.g. Curier et al., 2014; Vinken et al., 2014b; Terrenoire et al., 2015; Geddes et al., 2016; European Environment Agency, 2018). While the absolute contributions in these regions depend strongly on the emission inventory (5 to 10 $nmol\,mol^{-1}$), the relative values are generally 50 % and more. Accordingly, land transport emissions are one of the most important contributors to $NO_y$ in regions with large $NO_2$ concentrations.

These large amounts of $NO_x$ emissions from land transport clearly contribute to the formation of ozone, but the relative
contributions to ozone are lower than the contributions to $NO_y$. Here, the mean contributions range between 10 % and 16 % in most regions and even during extreme ozone events the contributions are below 30 %. Clearly, land transport emissions are an important contributor to European ozone levels, but they are not the most important contributor to European ozone levels. This is underlined by our analysis of the contribution of land transport emissions to ozone production in Europe, which range between 20 % to 40 % in most areas. The emission sectors which are most important for ozone production in Europe are
biogenic emissions and anthropogenic non-traffic emissions. During periods of large ozone values, however, our analyses show that the contribution of land transport emissions to ozone increases strongly, while the contribution of anthropogenic non-traffic emissions is only slightly changed. This suggests that emissions from land transport make an important contribution to large ozone values.

We find that the regions with the largest contribution of land transport emissions to ozone are not necessarily identical with
the regions with the largest contributions to reactive nitrogen. The ozone values peak mainly in Northern Italy (around the Po Valley) and Southern Germany, which is consistent with the findings of Tagaris et al. (2015). For the Po Valley in particular, ground-level measurements show that this is one of the regions with the largest ozone levels in Europe (e.g. Martilli et al., 2002; Guerreiro et al., 2014; European Environment Agency, 2018). In Southern England, around Paris, and the Benelux as well as Rhine-Ruhr regions, where the contribution of land transport emissions to $NO_y$ stands out, the contributions to ozone
are not the largest. The result, that regions are hot-spots for $NO_y$ from land transport emissions, but not for $O_3$ from land transport is counter intuitive. The reasons for this is that large amounts of $NO_x$ emissions alone are not sufficient for large ozone production. This is caused by the non-linearitiy of the ozone chemistry, including the availability of VOCs, and the strong interdependence of ozone production and meteorological conditions (e.g. Monks et al., 2015).

A detailed comparison of our results with previous studies is complicated: First, we apply one global tag for the land
transport sector and do not differentiate between local produced ozone and long range transported ozone. In comparison to our approach similar regional studies usually attribute ozone only to the emissions within the regional domain and attribute long-range transported ozone to the boundary conditions. Second, the methods for source apportionment applied in various studies differ. Third, the applied emission inventories differ, so do ozone metrics and simulated periods. Tagaris et al. (2015), who calculated the impact of different emission sectors on ozone using a 100 % perturbation of the respective emission sectors

reported an impact of European road transport emissions of 7 % on average for the maximum 8 hr ozone values in July 2006. In most regions impacts above 10 % were reported, with maximum local impacts (Southern Germany, Northern Italy) of above 20%. While their largest impacts occur in similar regions as our largest contributions (Southern Germany, Northern Italy), our mean contributions are larger than their impacts, but the maximum contributions are lower than their maximum impacts.

Further, around London and in parts of Northern England their impacts (see Fig. 3 therein) are around 2 to 4 %, while our contributions are in the range of 8 to 10 %. Hence, impact and contribution differ largely in these regions. This is in line with previous work, stating that for ozone source attribution contributions instead of impacts should be used (Grewe et al., 2012, 2019).

All the studies that we are aware of and which reported contributions of land transport emissions to ozone over Europe using

a tagging method either applied the CAMx model (CAMx OSAT method, Karamchandani et al., 2017) or the CMAQ model (CMAQ-ISM method, Valverde et al., 2016; Pay et al., 2019). As discussed, these two methods examine proxies of the ozone sensitivity to determine, whether ozone production is $NO_x$ or VOC limited. These previous studies considered only European emissions, while we consider the combined effect of European emissions and long range transport. Therefore, one would expect that our contribution analysis shows larger contributions compared to previous studies. However, our contributions in general

are lower compared to previously reported values. As an example, Karamchandani et al. (2017) reported contributions around larger European cities in the range of 11 to 24 %, in Budapest even up to 35 %. Valverde et al. (2016) reported contributions of road transport emissions from Madrid and Barcelona of up to 24 % and 8 %, respectively. Similarly, Pay et al. (2019) diagnosed contributions of road transport emissions on ozone of 9 % over the Mediterranean Sea and up to 18 % over the Iberian Peninsula, however for a specific summer episode only (July 2012). To discuss potential reasons why our contributions

are lower compared to previous estimates, we analysed our results for July 2010, to compare these contributions directly with the findings of Karamchandani et al. (2017).

As an example, Karamchandani et al. (2017) reported contributions of 17 % around Berlin, while our contributions are in the range of 12–14 %. Further they diagnosed contributions from the biogenic sector of around 11 % around Berlin, while we find contributions of the biogenic sector of around 18 %. Generally, the contributions reported by Karamchandani et al.

(2017) seem to be much more variable over Europe compared to our results. A reason for this might be the different treatment in the attribution of $NO_x$ and VOC precursors. Land transport emissions contribute mainly to $NO_x$ emissions, while biogenic emissions are an important source of VOCs. As shown by Butler et al. (2018), anthropogenic emissions contribute most to ozone over Europe, if a $NO_x$ tagging is applied, while biogenic emissions are the most important contributor, when a VOC tagging is applied (Figs. 3 and 4 therein). Accordingly, those approaches which use a threshold to perform either a VOC or

$NO_x$ tagging, attribute ozone production under VOC limitation mainly to biogenic sources, while under a $NO_x$ limitation ozone is attributed mainly to anthropogenic sources (including land transport emissions). Most likely this leads to a much stronger variability between anthropogenic and biogenic contributions compared to our approach, where ozone is always attributed to $NO_x$ and VOC or $HO_x$ precursors.

Similar effects can also be observed when comparing our results to the results of Lupaşcu and Butler (2019), who applied a

$NO_x$ tagging for the period April to September 2010 and considered regional as well as global sources similar to our approach.

They reported contributions of biogenic emissions in Europe for the period July - September between 5 and 13 % over Europe. Our results show contributions of biogenic emissions which are much larger (15 to 26 % for the same period). In their approach, ozone is only attributed to biogenic $NO_x$ emissions, while we attribute ozone to biogenic $NO_x$ and VOC emissions. In addition, our estimated stratospheric contribution to ground-level ozone is larger than the contributions reported by Lupaşcu and Butler (2019). Our results indicate stratospheric contributions for July to September in the range of 5 to 10 % compared to their 2 to 4 %. Similarly, for lightning-$NO_x$ our model shows larger contributions (6–12 %) compared to the 3–6 % diagnosed by Lupaşcu and Butler (2019).

The differences in the contributions for the stratospheric and the lightning category can partly be attributed to the more efficient vertical mixing in COSMO-CLM. Mertens et al. (2020) reported a maximum difference of the contributions from the stratosphere and lightning to ozone between EMAC and COSMO-CLM/MESSy of 30 %. The difference between our results for lightning and stratospheric contributions and the results of Lupaşcu and Butler (2019) are much larger as these 30 %. Therefore, the difference can most likely not be fully attributed to differences in vertical mixing. Rather, the differences can probably be explained by the different contributions of the biogenic category (due to different tagging methods) and by differences of lightning emissions and the treatment of stratospheric ozone.

In general, the various studies discussed above do not provide sufficient information on the emissions parametrisations and inventories used (e.g. for lightning-$NO_x$, soil $NO_x$ and biogenic VOCs) to fully analyse these differences. The discrepancy in the results of the different source attribution methods clearly shows that a coordinated comparison between these methods is important. This was already suggested by Butler et al. (2018).

The comparison of the results between the two emission inventories sheds light on the uncertainties associated with such a source attribution method. The differences of the results for the direct pollutants CO and $NO_y$ are rather large. The mean ozone contributions are much less influenced than the direct pollutants. Especially during winter and in the middle/upper troposphere the contributions are mainly dominated by long range transport (e.g. land transport emissions from the rest of the world). In our study, however, we focused only on the uncertainties caused by different emission inventories for Europe. Therefore, we did not investigate the influence of uncertainties from emissions from the rest of the world. Uncertainties of these emissions are likely to influence the contribution from long range transport.

The results based on the two emission inventories show only small differences between the mean ozone values, but large differences between the corresponding extreme ozone values and net ozone production rates, even though the total land transport emissions between the two emission inventories are similar. These differences are mainly due to the the differences between the anthropogenic non-traffic and the shipping emissions between the two emission inventories. Accordingly, the source attribution of land transport emissions is not only influenced by the uncertainties of the land transport emissions, but also by the uncertainties of all other emission sectors. Further, it is also likely that emission inventories underestimate land transport emissions. As an example, Kuik et al. (2018) reported an underestimation of road traffic emissions for Berlin by up to 50 %. The impact of such large underestimations on the source attribution results need to be investigated. Besides uncertainties of anthropogenic emissions, uncertainties of biogenic emissions also contribute to uncertainties of the source attribution results. In this context especially uncertainties of biogenic VOC emissions and $NO_x$ emissions from soil play an important role. As an

example, the uncertainties of soil-$NO_x$ are rather large (Vinken et al., 2014a) and the emissions applied in our model system are at the lower end of current emission estimates. Similar large uncertainties are also reported for biogenic VOC emission inventories (Ashworth et al., 2010; Han et al., 2013; Oderbolz et al., 2013).

Generally, uncertainties caused by the emissions are larger than the uncertainties, which are caused by the simplifications applied in our source attribution method, which are in the order of some percent (see also discussion by Mertens et al., 2018). Further, our results indicate a large seasonal variability of the contribution of land transport emissions to ozone. This variability is not only caused by the meteorological conditions but also by the seasonal cycle of other emissions. Accordingly, not only the total emissions of different emission sectors but also their seasonality (and the correct representation of this seasonality) plays an important role.

The question of how to evaluate these source attribution results also remains a challenge. Clearly, a comparison of different source attribution methods would help in revealing individual strengths and weaknesses of the methods. In addition, measurements of specific episodes or in specific regions (e.g. in plumes of cities, in regions with strong lightning activity or events of stratospheric intrusions) can help to assess the diagnosed contributions by investigating, if these contributions are in a plausible range. Further, the influence of model biases on the analysed contributions can be estimated, but a direct evaluation of these contributions is not possible. However, the diagnostic information from the source attribution methods can help to understand the modelled ozone production in more detail and can offer important insights to understand potential model biases.

## 8    Conclusions

In the present study we investigate the contributions of land transport emissions to pollutants in Europe and Germany. focusing on ozone ($O_3$), carbon monoxide ($CO$) and reactive nitrogen ($NO_y$) by means of simulations with the MECO(n) model system. This model system couples a global chemistry-climate model on-line with a regional chemistry-climate model. To quantify the contributions of land transport emissions to these species we used a tagging method for source attribution. This tagging method is an accounting system, which completely decomposes the budgets of ozone and ozone precursors into contributions from different emission sources. For the first time such a method is applied consistently in the global as well as the regional models to attribute ozone and ozone precursors to the emissions of land transport. To consider the uncertainties associated with the emission inventories, we performed simulations with two different emission inventories for Europe.

The contribution of land transport emissions to ground-level $NO_y$ depends strongly on the applied emission inventory. In general the contributions range from 5 to 10 $\mathrm{nmol\,mol^{-1}}$ near the European hot-spot regions, which are Western and Southern Germany, the Po Valley, Southern England as well as the Paris and Moscow metropolitan region. In most other parts in Central and Southern Europe contributions of around 2 to 3 $\mathrm{nmol\,mol^{-1}}$ are simulated. Generally, absolute contributions during winter are larger than during summer, but the seasonal differences are smaller than the differences between different emission inventories. The absolute contributions correspond to relative contributions of 50 to 70 % to ground-level $NO_y$, which indicates that land transport emissions are one of the most important sources for $NO_y$ near ground-level.

Similar as for $NO_y$ the simulated contribution of land transport emissions to CO near ground-level depends strongly on the applied emission inventory. Generally, the contributions range around $30\,\text{nmol}\,\text{mol}^{-1}$ during summer in regions which are not directly associated with large land transport emission sources and more than $75\,\text{nmol}\,\text{mol}^{-1}$ near emission hot-spots such as Paris or Moscow.

The contribution of land transport emissions to ozone, which is a secondary pollutant, shows a geographical distribution which differs strongly from the distribution of the primary emissions. The absolute contribution shows a strong North-West South-East gradient with the largest contributions around the Mediterranean Sea. Due to the non-linear behaviour of ozone chemistry and the strong dependency of ozone formation on the meteorology and other precursors as $NO_x$ (mainly CO, $CH_4$ and VOCs), regions with large emissions in Western Europe (Benelux, British Islands, Western Germany) show no peak of

the contribution of land transport emissions to ozone. Such a peak is simulated in the Po Valley, where large emissions and favourable conditions for ozone production prevail. Generally, the contribution has a strong seasonal cycle with values of 2 to $3\,\text{nmol}\,\text{mol}^{-1}$ during winter and 5 to $10\,\text{nmol}\,\text{mol}^{-1}$ during summer. These absolute contributions correspond to relative contributions in the range of 8 to 16 %. During winter, the results obtained for the two European emission inventories show almost no differences. The contributions are largely determined by long range transport and the year-to-year variability is the

largest source of uncertainty. Of course, also the uncertainties in the emission inventories for emissions outside of Europe can influence the contribution analyses noticeably, but this has not been investigated in the present study. During summer the differences between the contributions diagnosed using the two emission inventories are larger than the year-to-year variability. Hence, during summer uncertainties of emission inventories for Europe influence the contribution analyses considerably.

     While the emissions of the land transport sector have almost no seasonal cycle, the contributions exhibit a strong seasonal

cycle. This shows the strong influence of seasonal cycles of other emission sources on the ozone production from land transport emissions. Hence, uncertainties of total emissions, geographical distributions and the seasonal cycles of other emissions strongly influence the contribution analysis of land transport emissions. Especially during summer biogenic emissions play a key role here. The impact of uncertainties of these emissions needs to be studied in more detail. In addition, the impact of the applied source attribution method needs to be investigated in a coordinated way. Our results suggest, that our methodology,

which accounts for $NO_x$ and VOCs at the same time, leads to a partitioning between anthropogenic and biogenic sources partly different from previous studies which account for either $NO_x$ or VOCs.

     The contribution of land transport emissions to extreme (99th percentile) ozone values is largest in the Po Valley, reaching up to roughly 28 %. In other regions of Europe the contribution of land transport emissions to extreme ozone events is lower and strongly depends on the region and the emission inventory. Important is, however, that the contribution of land transport

emissions to ozone increase with increasing ozone levels. This indicates that land transport emissions play an important role for high ozone events. Generally, the contribution of land transport emissions to ozone production up to $850\,\text{hPa}$ is around 20 and 40 % in most European regions. However, only in very few regions land transport emissions are the most important contributor to the ozone production. In most regions anthropogenic non-traffic and biogenic emissions are more important. Our analysis shows that especially also the biogenic emissions are important during high ozone events. Their contribution increases with

increasing ozone levels similar to the contribution of land transport emissions. The contribution of anthropogenic non-traffic

emissions shows almost no increase. However, the large differences obtained for the two emission inventories indicate a large uncertainty range of such analysis.

As discussed in our introduction (see Table 1) contributions, provide the share of ozone caused by specific emissions. With respect to mitigation options it is important to point out that these contributions provide no information about how ozone changes if the corresponding emission sources are reduced. As discussed, this question can only be assessed with the perturbation approach. However, the contributions indicate how important different emission sources are for the tropospheric ozone budget in Europe.

As a next step the analysis will be refined using source attribution categories, which differentiates between contributions of European land transport emissions and land transport emissions from the rest of the world. Such an analysis will help to quantify the importance of European and global land transport emissions to ozone levels in Europe. Further, more reliable emission estimates are important for follow up studies. Here, the focus should not only be on the land transport emissions, but also on other important emissions, including especially biogenic VOCs and soil-$NO_x$ emissions, which are subject to large uncertainties and contribute strongly to European ozone levels. To better constrain the uncertainties of the contribution analysis follow up studies are planned (see Sect. 7) in which we will combine observational data of specific aircraft measurement campaigns together with model results including the analysed contributions.

*Code and data availability.* The Modular Earth Submodel System (MESSy) is continuously further developed and applied by a consortium of institutions. The usage of MESSy and access to the source code is licenced to all affiliates of institutions which are members of the MESSy Consortium. Institutions can become a member of the MESSy Consortium by signing the MESSy Memorandum of Understanding. More information, including on how to become licensee for the required third party software, can be found on the MESSy Consortium Website (http://www.messy-interface.org). The simulations have been performed with a release of MESSy based on version 2.50. All changes are available in the official release (version 2.51). The namelist set-up used for the simulations is part of the electronic supplement. The model data used in this study are available under the following DOI: (The DOI needs to be added at a later stage, because the SuperMUC is currently not available due to hacker attack)

*Author contributions.* MM performed the simulations, analysed the data and drafted the manuscript. AK and PJ developed the model system. VG developed the tagging method. RS drafted the study. All authors contributed to the interpretation of the results and to the text.

*Competing interests.* The authors declare that they have no competing interests.

*Acknowledgements.* M. Mertens acknowledges funding by the DLR projects 'Verkehr in Europa' and 'Auswirkungen von $NO_x$'. Furthermore, part of this work is funded by the DLR project 'VEU2'. A. Kerkweg acknowledges funding by the German Ministry of Edu-

cation and Research (BMBF) in the framework of the MiKlip (Mittelfristige Klimaprognose/Decadal Prediction) subproject FLAGSHIP (Feedback of a Limited-Area model to the Global-Scale implemented for HIndcasts and Projections, funding ID 01LP1127A) and in the framework of CMIP6, subproject CMIP6-Chemie-TP2, funding ID 01LP1606B. Analysis and graphics of the used data was performed using the NCAR Command Language (Version 6.4.0) Software developed by UCAR/NCAR/CISL/TDD and available on-line: http://dx.doi.org/10.5065/D6WD3XH5. We thank Helmut Ziereis (DLR) and Markus Kilian (DLR) for very valuable comments improving the manuscript. Further, we would like to thank two anonymous referees who helped to improve the quality of the manuscript. We acknowledge the Leibniz-Rechenzentrum in Garching for providing computational resources on the SuperMUC2 under the project id PR94RI.

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

**Table 1.** Comparison of scientific questions which can be answered by impacts (using a perturbation method) and contributions (calculated by a source attribution method such as tagging).

| Questions | impacts (perturbation) | contribution (attribution) |
| --- | --- | --- |
| Which ozone concentration can be attributed to a specific emission source? What is the share of individual emissions of the ozone budget? | not suitable | suitable |
| Which anthropogenic source has the largest contribution to ozone? | not suitable, as part of ozone remains unexplained |  well suited as 100 % of ozone can be explained |
| What source should be taken into account for mitigation options, because it has the largest ozone share? | not suitable | suitable |
| How does the share in ozone of an emission change, if the strength of that emission is changed? | not suitable | suitable |
| What source should be taken into account for mitigation options, because it decreases ozone concentration most? | suitable | not suitable |
| What is the resulting ozone change, because of a change in the strength of emission? |  well suited, as sensitivity of ozone on the emission change is analysed | not suitable |
| Can the quantity be measured? | yes, effects of emission reductions can be measured and compared with model results. | no, so far no measurement concept is available. |
| How large (and for which sources) are compensating/feedback effects caused by a change in the strength of an emission? | combination of both methods required | |

**Table 2.** Overview of the most important MESSy submodels applied in EMAC and COSMO/MESSy, respectively. Both COSMO/MESSy instances use the same set of submodels. MMD* comprises the MMD2WAY submodel and the MMD library.

| Submodel | EMAC | COSMO | short description | references |
|---|---|---|---|---|
| AEROPT | x | | calculation of aerosol optical properties | Dietmüller et al. (2016) |
| AIRSEA | x | x | exchange of tracers between air and sea | Pozzer et al. (2006) |
| CH4 | x | | methane oxidation and feedback to hydrological cycle | |
| CLOUD | x | | cloud parametrisation | Roeckner et al. (2006), Jöckel et al. (2006) |
| CLOUDOPT | x | | cloud optical properties | Dietmüller et al. (2016) |
| CONVECT | x | | convection parametrisation | Tost et al. (2006b) |
| CVTRANS | x | x | convective tracer transport | Tost et al. (2010) |
| DDEP | x | x | dry deposition of aerosols and gas phase tracers | Kerkweg et al. (2006a) |
| EC2COSMO | x | | additional ECHAM5 fields for COSMO coupling | Kerkweg and Jöckel (2012b) |
| GWAVE | x | | parametrisation of non-orographic gravity waves | Roeckner et al. (2003) |
| JVAL | x | x | calculation of photolysis rates | Landgraf and Crutzen (1998), Jöckel et al. (2006) |
| LNOX | x | | $NO_x$-production by lighting | Tost et al. (2007), Jöckel et al. (2010) |
| MECCA | x | x | tropospheric and stratospheric gas-phase chemistry | Sander et al. (2011), Jöckel et al. (2010) |
| MMD* | x | x | coupling of EMAC and COSMO/MESSy (i.e. library and submodel) | Kerkweg and Jöckel (2012b); Kerkweg et al. (2018) |
| MSBM | x | x | multiphase chemistry of the stratosphere | Jöckel et al. (2010) |
| OFFEMIS | x | x | prescribed emissions of trace gases and aerosols | Kerkweg et al. (2006b) |
| ONEMIS | x | x | on-line calculated emissions of trace gases and aerosols | Kerkweg et al. (2006b) |
| ORBIT | x | x | Earth orbit calculations | Dietmüller et al. (2016) |
| QBO | x | | Newtonian relaxation of the quasi-biennial oscillation (QBO) | Giorgetta and Bengtsson (1999), Jöckel et al. (2006) |
| RAD | x | | radiative transfer calculations | Dietmüller et al. (2016) |
| SCAV | x | x | wet deposition and scavenging of trace gases and aerosols | Tost et al. (2006a) |
| SEDI | x | x | sedimentation of aerosols | Kerkweg et al. (2006a) |
| SORBIT | x | x | sampling along sun synchronous satellite orbits | Jöckel et al. (2010) |
| SURFACE | x | | surface properties | Jöckel et al. (2016) |
| TAGGING | x | x | source attribution using a tagging method | Grewe et al. (2017) |
| TNUDGE | x | x | Newtonian relaxation of tracers | Kerkweg et al. (2006b) |
| TROPOP | x | x | diagnostic calculation of tropopause height and additional diagnostics | Jöckel et al. (2006) |

**Table 3.** Description of the different tagging categories applied in this study following Grewe et al. (2017). Please note that some tagging categories summarise different emission sectors (see description). The last row shows the nomenclature of the tagged tracers exemplary for ozone.

| tagging category | description | notation for tagged ozone |
|---|---|---|
| land transport | emissions of road traffic, inland navigation, railways (IPCC codes 1A3b_c_e) | $O_3^{tra}$ |
| anthropogenic non-traffic | sectors energy, solvents, waste, industries, residential, agriculture | $O_3^{ind}$ |
| ship | emissions from ships (IPCC code 1A3d) | $O_3^{shp}$ |
| aviation | emissions from aircraft | $O_3^{air}$ |
| lightning | lightning-$NO_x$ emissions | $O_3^{lig}$ |
| biogenic | on-line calculated isoprene and soil-$NO_x$ emissions, off-line emissions from biogenic sources and agricultural waste burning (IPCC code 4F) | $O_3^{soi}$ |
| biomass burning | biomass burning emissions | $O_3^{bio}$ |
| $CH_4$ | degradation of $CH_4$ | $O_3^{CH4}$ |
| $N_2O$ | degradation of $N_2O$ | $O_3^{N2O}$ |
| stratosphere | downward transport from the stratosphere | $O_3^{str}$ |

**Table 4.** Definition of the chemical families used in the tagging method. More details on the species contained in the families are given in the Supplement of Grewe et al. (2017).

| Tagged species | Description |
|---|---|
| $O_3$ | Ozone as family of odd oxygen |
| PAN | PAN |
| CO | CO |
| NOy | all chemically active nitrogen compounds without PAN in the chemical mechanisms (15) |
| NMHC | all NMHCs in the chemical mechanisms (42) |
| OH | OH tagged in a steady state approach (see Rieger et al., 2018) |
| $HO_2$ | $HO_2$ tagged in a steady state approach |

**Table 5.** Average (2008 to 2010) annual total emissions for the CM50 domain of different anthropogenic emission sectors and the total of all emission sectors for $NO_x$ (in $Tg(NO)\ a^{-1}$), CO ($Tg(CO)\ a^{-1}$), VOC ($Tg(C)\ a^{-1}$) and the $NO_x$ to VOC ratio ($NO_x$/VOC)).

| emission sector | *REF* | | | | *EVEU* | | | |
| --- | --- | --- | --- | --- | --- | --- | --- | --- |
| | $NO_x$ | CO | VOC | $NO_x$/VOC | $NO_x$ | CO | VOC | $NO_x$/VOC |
| land transport | 5.2 | 29 | 3.1 | 1.7 | 5.4 | 24 | 3.4 | 1.6 |
| anthropogenic non traffic | 7.3 | 28 | 14 | 0.52 | 5.1 | 30 | 6.5 | 0.78 |
| shipping | 2.4 | 0.25 | 0.36 | 6.5 | 1.8 | 0.30 | 0.096 | 19 |
| aviation | 0.60 | - | - | - | 0.55 | - | - | - |
| Total | 15.5 | 57.3 | 17.5 | 0.88 | 12.9 | 54.3 | 10.0 | 1.3 |

**Table 6.** Average (2008–2010) annual total emissions for the CM50 domain of $NO_x$ (in $Tg(NO)\,a^{-1}$), CO ($Tg(CO)\,a^{-1}$), VOC ($Tg(C)\,a^{-1}$) and the $NO_x$ to VOC ratio ($NO_x/VOC$). Given are the total emissions of the emission sectors which are identical in *REF* and *EVEU*.

| emission sector | $NO_x$ | CO | VOC | $NO_x/VOC$ |
|---|---|---|---|---|
| biogenic | 1.2 | 4.8 | 22 | 0.056 |
| biomass burning | 0.26 | 9.0 | 0.377 | 0.73 |
| agricultural waste burning | 0.081 | 2.845 | 0.0981 | 0.83 |
| lightning | 0.76 | - | - | - |

**Table 7.** Root-mean-square error (RMSE, in $\mu/\text{gm}^3$) and mean bias (MB, in percent) of the *REF* and *EVEU* simulations compared to Airbase v8 observation data. Given are the scores for the mean values during JJA and DJF, as well as values for 95th percentile for JJA. For *REF* listed additionally also the scores considering only the values at 12 and 15 UTC.

|  | RMSE ($\mu/\text{gm}^3$) | MB (%) |
|---|---|---|
| REF JJA mean | 29.2 | 26.6 |
| REF JJA 12 and 15 UTC | 18.7 | 13.4 |
| EVEU JJA mean | 24.3 | 20.5 |
| REF JJA 95th percentile | 26.9 | -10.0 |
| EVEU JJA 95th percentile | 28.7 | -14.2 |
| REF DJF mean | 35.1 | 32.8 |
| EVEU DJF mean | 32.8 | 30.1 |

**Table 8.** Contribution of different emission sources area averaged over Europe (defined as rectangular box $10°$ W to $38°$ E and $30°$ N to $70°$ E) for JJA 2008–2010 at three different altitudes (in %). The values are mean values of the *REF* and *EVEU* simulation, the range indicates the standard deviation between the results of *REF* and *EVEU*.

|  | ground (%) | 600 hPa (%) | 200 hPa (%) |
|---|---|---|---|
| stratosphere | $7.4 \pm 0.1$ | $13.7 \pm 0.1$ | $52.0 \pm 0.1$ |
| $CH_4$ | $14.3 \pm 0.1$ | $14.7 \pm 0.1$ | $8.3 \pm 0.1$ |
| lightning | $8.8 \pm 0.2$ | $15.0 \pm 0.5$ | $9.0 \pm 0.1$ |
| aviation | $3.7 \pm 0.1$ | $5.2 \pm 0.1$ | $2.0 \pm 0.1$ |
| biomass burning | $6.1 \pm 0.1$ | $4.8 \pm 0.1$ | $2.2 \pm 0.1$ |
| biogenic | $18.8 \pm 0.3$ | $15.7 \pm 0.1$ | $7.5 \pm 0.1$ |
| shipping | $9.2 \pm 0.6$ | $4.7 \pm 0.1$ | $1.5 \pm 0.1$ |
| anth. non-traffic | $16.4 \pm 0.8$ | $13.0 \pm 0.2$ | $6.1 \pm 0.1$ |
| land transport | $11.6 \pm 0.4$ | $8.3 \pm 0.1$ | $3.3 \pm 0.1$ |
| $N_2O$ | $3.6 \pm 0.1$ | $5.1 \pm 0.0$ | $8.3 \pm 0.1$ |

**Table 9.** Diagnosed net ozone production ($P_{O3}$) of the ten considered categories (in Tg a$^{-1}$) as simulated by CM50. The production rates are integrated over the CM50 domain and up to 850/200 hPa, respectively. The values are averaged for 2008–2010, the ranges indicate one standard deviation with respect to time based on the annual averages of the individual years.

| | $P_{O3}$ integrated up to 850 hpa (Tg a$^{-1}$) | | $P_{O3}$ integrated up to 200 hpa (Tg a$^{-1}$) | |
| --- | --- | --- | --- | --- |
| | REF | EVEU | REF | EVEU |
| land transport | $13.2 \pm 0.2$ | $13.3 \pm 0.3$ | $22.8 \pm 0.6$ | $23.4 \pm 0.5$ |
| anthropogenic non-traffic | $22.2 \pm 0.5$ | $15.1 \pm 0.3$ | $37.8 \pm 1.1$ | $26 \pm 0.5$ |
| shipping | $6.7 \pm 0.1$ | $5.6 \pm 0.1$ | $10.6 \pm 0.1$ | $8.8 \pm 0.1$ |
| aviation | $0.3 \pm 0.1$ | $0.1 \pm 0.1$ | $8.1 \pm 0.1$ | $7.9 \pm 0.1$ |
| biogenic | $15.9 \pm 0.6$ | $15.3 \pm 0.5$ | $28.8 \pm 0.7$ | $28.2 \pm 0.7$ |
| lightning | $-0.9 \pm 0.1$ | $-1.0 \pm 0.1$ | $6.9 \pm 0.3$ | $7.0 \pm 0.3$ |
| biomass burning | $2.1 \pm 0.2$ | $1.8 \pm 0.1$ | $3.8 \pm 0.3$ | $3.5 \pm 0.3$ |
| CH$_4$ degradation | $4.5 \pm 0.1$ | $3.6 \pm 0.1$ | $12.5 \pm 0.4$ | $11.5 \pm 0.4$ |
| N$_2$O | $-0.2 \pm 0.1$ | $-0.3 \pm 0.1$ | $1.8 \pm 0.1$ | $1.7 \pm 0.1$ |
| stratosphere | $-1.9 \pm 0.1$ | $-1.7 \pm 0.6$ | $-10.9 \pm 0.7$ | $-11 \pm 0.7$ |
| total | $61.8 \pm 0.3$ | $51.9 \pm 1.0$ | $122.3 \pm 2.0$ | $107.4 \pm 1.8$ |