# Peer review of "Attributing ozone and its precursors to land transport emissions in Europe and Germany"

_Atmospheric Chemistry and Physics, 2019_

## Referee Comment (RC1) · Anonymous Referee #1 · 8 Nov 2019

This publication presents an analysis of the role of transport emissions on different pollutants by using a tagging source apportionment approach. The uncertainties related to the use of different emission inventories are also assessed. The paper is well structured although the English should be reviewed. In this respect, I listed some possible improvements (see the minor comment section) but the whole text would need to be revised. Although I find this work of interest, I listed below some major concerns I have regarding the methodology proposed by the Authors and would appreciate some additional information in the text regarding these points before I could recommend publication.

Major comments:

1. As noted by the Authors in their introduction, sensitivity analysis and tagging approach are two approaches that are used to answer different questions. Sensitivity deliver impacts whereas tagging delivers contributions. While it is rather clear that impacts can be used to inform on the potential effects of emission reductions on air quality levels, it is rather unclear how the contributions estimated for the Authors can be used in practice. In one of their earlier work, the Authors mentioned the possibility of using contributions in complement to the impacts to inform on the potential of emission reductions that go beyond the threshold covered by the perturbation or sensitivity method. But this possibility is not mentioned in this work. On the contrary, confusion is introduced in some sections in which the Author seem to indicate that contributions can be used to support air quality strategies, e.g. in Section 4.2 (first three lines).

2. Along the same lines, the Authors mention that these findings based on tagging are in line with other studies using perturbation methods (see l27 in the discussion section). How can these conclusions be reached when it is clearly mentioned in the introduction that perturbation methods and tagging are expected deliver different results. These two statements contradict each other, unless O3 may be considered as a linear species, in which case both methods would indeed converge to the same conclusions

3. In some sections, many numbers are given to characterize the various contributions, e.g. Section 5. A few additional lines to detail the implication these results may have would be useful.

Minor comments:

Most of these comments address spelling errors or unclear grammatical sentences. But I would strongly suggest the Authors to review the whole text regarding the English writing.

1. In many sentences "as" is used in place of "than" (e.g. p19 l27; p22 l6; p26 l25. . .) 2. P1 l28: teh –> the 3. P3 l20: quantifies –> quantified 4. P4 l29: Th –> the 5. P5 l34: an –> a 6. P5 l35: to –> too 7. P8 l9: not –> note 8. P8 l8: party –> part 9. P19 l16: kept –> be kept 10. P19 l 25: the text within parentheses is unclear 11. P19 l29:

increase –> increases 12. P19 l34: all most –> almost 13. P22 l14 & 15: sentence is unclear 14. P23 l3: second –> second most 15. P23 l11: is displayed –> are displayed 16. Discussion section: could the Authors add a few words to explain how all these contribution numbers can be validated? Can we use contributions to know which inventory might be closer to the truth? 17. P25 l3: corresponds –> correspond 18. P25 l7: depend –> depends 19. P25 l8: contributor –> contributors 20. P25 l17: increase –> increases 21. P25 l19: regions of –> regions with 22. P25 l25: not largest –> not the largest 23. P27 l11: by different –> between different 24. P27: l28 to 30: please check the use of the word "uncertainty" which is used many times in a couple of sentences 25. P28 l2: studies –> studied 26. P28 l9: o–> ? 27. P28 l9 region –> regions 28. P28 l11 increase –> increases 29. P28 l19-20: can the Author develop a little bit more on how they plan to use observation data to validate the contributions? I believe this is a key point and one of the major benefits of the tagging approach.

---

## Referee Comment (RC2) · Anonymous Referee #2 · 13 Nov 2019

Mertens et al. perform a source attribution study examining the contribution of different emission sectors to air pollution over Europe, with a focus on ozone as a pollutant, a special focus on emissions from the road transport sector, and a regional focus on Europe and Germany. They employ a uniform methodology for "tagging" the emissions of ozone precursors in a system of coupled models, allowing a consistent downscaling to be made from the global scale to the national scale. Furthermore, they compare two simulations performed with different emission inventories, showing the sensitivity of the sectoral contributions to the way in which the emissions from each sector are represented in the emission inventory. This combination of sensitivity and source attribution reveals some interesting information about the behaviour of of tropospheric ozone in the model system used, for example the particularly strong differences in the contribution of land transport emissions to the higher percentiles of the ozone distribution when a spatially more explicit inventory is used.

The manuscript is clearly within the scope of ACP, and the method clearly has a lot of potential to inform international air quality policy. Unfortunately the manuscript in its current form suffers from a number of serious flaws, which must be corrected before it can be accepted for publication.

Firstly, the quality of the written English is terrible. The manuscript is littered with grammatical and spelling errors, and written in a generally inaccessible style. I do not feel that it is my job as a reviewer to provide an exhaustive list of these errors. The authors should seek additional help to get the language up to an acceptable standard. I will give one example though: the very title of the manuscript contains a jarring error. The current title basically implies that ozone causes land transport emissions. Clearly this is the other way around. Land transport emissions happen first, and this leads to ozone production. A grammatically correct title could be "Attributing ozone and its precursors to land transport emissions in Europe and Germany".

Specific Comments

In the abstract, the authors state that tagging is "required" and that their method is the "only possible" way to examine global to regional scale effects. This language is way too strong and should be toned down before publication. This is especially true given that the authors themselves state on line 28 of page 25 that their results are "consistent" with a perturbation study, and also given the fact that the experiment design doesn't actually make a distinction between land transport emissions in Europe and the rest of the world.

Page 2, lines 26-27: while this is generally true on very small scales (eg. urban areas), the response of ozone to perturbation of precursor emissions in remote regions has been shown to be approximately linear. See for example Wild et al. (2012) and Turnock et al. (2018). Since the authors are also discussing long-range transport,

some additional discussion of this here would be relevant.

Page 4, lines 3-4: Aren't the last two points in this list in fact exactly the same thing?

Page 4, line 5: I can see how using two different inventories can somewhat account for uncertainties in the emissions, but three years is way to short a period to account for interannual variability. I also do not see how the model uncertainty or the uncertainty in the choice of source apportionment method is accounted for at all in this experiment design. It's fine to mention that there can be a lot of uncertainty, but the authors should not claim to be doing more to address these uncertainties than they actually are.

Page 5, lines 32-35: These are the only lines in the paper where the authors discuss model evaluation. I understand that the model has been evaluated elsewhere, and the model is basically as {good,bad} as other models, but I would appreciate some more discussion about how the model performance could be expected to influence the conclusions of the manuscript. Since the authors also want to use their model to examine extreme ozone events (in Section 4.2), there must be at least some analysis of how well the model is capable of representing these events in comparison with observations.

Section 2.1: the authors need to do a lot more here to compare their source apportionment method with other methods in the literature. This is especially important, since the authors themselves have stated on Page 4 (line 1) that differences between source apportionment methods are an important source of uncertainty. Kwok et al. (2015) is already mentioned in Section 2.1, and Dunker et al. (2002) is mentioned in the introduction. Both of these studies use a regime-dependent attribution methodology, which is actually correctly acknowledged by the authors on page 26 in the Discussion section, but a discussion of how these methodologies differ from the methodology employed by the authors, and how this could be expected to influence the results of the study is required already in Section 2.1. Similarly, since the authors are also considering the global scale, they should also put their methodology into the context of the existing techniques for source attribution at the global scale. The authors already cite Emmons

et al. (2012) elsewhere in the paper, but do not mention this work in Section 2.1, where it would be appropriate to have some discussion of how these methods differ, and how this might influence the results of the study. One very important difference is that Emmons et al. (2012) only consider NOx as a precursor of ozone, while the technique employed by the authors combines the effects of both NOx and VOC precursors. Similarly, the study of Butler et al. (2018) is also missing from the discussion. Butler et al. (2018) account for effects of both NOx and VOC as ozone precursors, but they make some very different design decisions to the technique employed by the authors. The authors must do more to put their method in the context of the previous work, and discuss the relative strengths and weaknesses of the approach they have chosen.

Also in Section 2.1, the authors could briefly mention how stratospheric ozone is tagged in their approach, since this does not fit into the framework of their Equation 2.

Section 2.2: The authors should make it clear that the tags are applied globally, with no distinction between emissions in Europe and the rest of the world. This is acknowledged later in the manuscript, but the reader would benefit from having this made clear already in this section.

Page 10, lines 17-24: For some additional context here, it would be nice to know how the proportional contributions of land transport to ambient modelled NOy compare to the proportional contribution of land transport to total NOx in the inventories. Is the contribution as would be expected from simply looking at the emissions, or is it disproportionally higher or lower?

Section 4.1, page 15: The authors rightly interpret the ozone due to land transport in DJF as coming from long-range transport. I also understand that the limits of the experimental design (one global tag for land transport) make it hard to say anything about long-range transport in JJA, when local photochemistry is more active. But could it be possible to try? For example, could they look at the the land transport contribution at the western boundary of the refined grid in JJA, and use this as a rough estimate

of the contribution of land transport (and other sectors) in remote regions to baseline ozone in Europe? This could add a lot of value to the study and would be highly relevant for international policymaking.

Page 16, line 3: the seasonal cycle of photochemical activity also plays a role here.

Page 16, line 8: is there any way in this study to separate the influence of soil NOx and biogenic VOC? Or are these two different sources inextricably joined together into the "biogenic" sector?

Section 4.2: As mentioned earlier, it would be nice to know how well the model is capable of reproducing the extreme values of ozone as measured. If the model is doing a good job at this, then the results reported here could help to understand these extreme ozone measurements. If the model is not doing well at this, then the results reported here could potentially provide information about systematic model biases, and point the way towards improving the model. As it currently stands, it is not clear at all how these results should be interpreted.

Page 19, line 11: The region "Alps" includes the Po Valley. Does this mean that high mountains are in the same region as a polluted valley? The influences on air quality would be expected to be very different in these regions. High mountains will be more influenced by the free troposphere (and long-range transport), while the valley will be more influenced by local sources. Furthermore, "Alps" and "Po Valley" are used individually in this section and elsewhere in the manuscript. It is not always clear which region is meant. The authors could consider disaggregating this region into two sub-regions for their analysis (which could be quite informative), or at least being clearer about exactly which region they are referring to throughout the text.

Page 19, lines 15-16: the discussion about "uncertainties" in the inventory is very vague here. Could the large range in the contribution of land transport to extreme ozone when using EVEU emissions be related to the higher spatial heterogeneity and existence of more "hot spots" in this inventory compared with REF? There could potentially be some important information here about the need to get the distribution of NOx right in order to capture the high ozone events. A comparison of the REF and EVEU ozone timeseries with some measurements from urban background stations during extreme events could potentially add a lot of value here.

Page 23, line 4: the results are not "rather similar", but actually have some important differences, which are subsequently discussed. I think what the authors are trying to say here is that the contribution of land transport is similar in each case, but this is not the meaning which comes across.

Page 25, lines 25-26: This sentence basically conveys no meaning and could be easily deleted with no loss to the manuscript. Alternatively the authors could try to be clearer about what they mean here.

Page 25, last paragraph: if the previous work only accounts for the contribution of European land transport emissions to European ozone, and the current study also includes global emissions, then shouldn't the current study result in a higher contribution than the previous work? The opposite appears to be the case. Can the authors explain this apparent discrepancy?

Page 26, line 23: the authors appear to be concluding from the strong influence of the "biogenic" sector that soil NOx emissions are strongly influencing ozone. But couldn't this also be biogenic VOC? How do they separate the influence of these two different sources? A comparison with Butler et al. (2018) could be instructive here, since in that study the separate roles of NOx and VOC as ozone precursors were examined. Comparison of their Figure 3 and Figure 4 indicates that biogenic VOC make a larger contribution to European ozone in summer than biogenic NOx. The authors should discuss this here.

Page 28, lines 14-15: the future work proposed by the authors would indeed be extremely interesting from a policymaking perspective. If possible, they should also include as many other sectors as possible. This could help to inform decisions about

where emission reductions would be most effective.

Page 28, lines 28-29: again, it appears that the authors are over-interpreting their results when they conclude that soil NOx has a strong influence on European ozone levels.

Additional References

Butler, T., Lupascu, A., Coates, J., and Zhu, S.: TOAST 1.0: Tropospheric Ozone Attribution of Sources with Tagging for CESM 1.2.2, Geosci. Model Dev., 11, 2825–2840, https://doi.org/10.5194/gmd-11-2825-2018, 2018.

Turnock, S. T., Wild, O., Dentener, F. J., Davila, Y., Emmons, L. K., Flemming, J., Folberth, G. A., Henze, D. K., Jonson, J. E., Keating, T. J., Kengo, S., Lin, M., Lund, M., Tilmes, S., and O'Connor, F. M.: The impact of future emission policies on tropospheric ozone using a parameterised approach, Atmos. Chem. Phys., 18, 8953–8978, https://doi.org/10.5194/acp-18-8953-2018, 2018.

Wild, O., Fiore, A. M., Shindell, D. T., Doherty, R. M., Collins, W. J., Dentener, F. J., Schultz, M. G., Gong, S., MacKenzie, I. A., Zeng, G., Hess, P., Duncan, B. N., Bergmann, D. J., Szopa, S., Jonson, J. E., Keating, T. J., and Zuber, A.: Modelling future changes in surface ozone: a parameterized approach, Atmos. Chem. Phys., 12, 2037-2054, https://doi.org/10.5194/acp-12-2037-2012, 2012.

---

## Author Response (AR1)

Dear editor

Thank you very much for guiding trough the editorial process.

According to the referee comments we thoroughly revised our manuscript.

The most important changes are:

- We added a additional discussion about the perturbation and the tagging approaches in the Introduction.
- We revised Section 2.1 and included a comparison of our tagging method with other methods.
- We added a Section 2.3 containing a brief comparison of the model results with observational data.
- We added an additional figure in Sect. 4.1 to discuss also the 'inflow' towards Europe.
- We divided the region 'Alps' in two sub-regions ('Northern Alps' and 'Po Valley') and adjusted Sect. 4.2 accordingly.
- We revised the Discussion completely.

Furthermore, we have carefully checked the language of the revised manuscript and clarified many issues. This leads to a huge amount of (small) changes in different parts of the revised manuscript. Please note, that we performed one last language check after uploading the replies to referee#1 and referee#2. Therefore, some changes of the revised manuscript differ slightly (wr.t. to the language) from our initial replies. We updated these changes in the attached replies to the referees (given in bold).

Attached are the comments to the two referees (original comments in italic, answers in normal fonts, changes in the revised manuscript in bold) together with the revised manuscript. In the revised manuscript all modifications are highlighted (latexdiff).

We are looking forward to your reply,

Mariano Mertens
(on behalf of all co-authors)

Dear referee#1

Thank you very much for your review of our manuscript acp-2019-715. Please find our replies to your comments below. In the following, referee comments are given in italics, our replies in normal font, and text passages which we included in the text are in bold.

*This publication presents an analysis of the role of transport emissions on different pollutants by using a tagging source apportionment approach. The uncertainties related to the use of different emission inventories are also assessed. The paper is well structured although the English should be reviewed. In this respect, I listed some possible improvements (see the minor comment section) but the whole text would need to be revised. Although I find this work of interest, I listed below some major concerns I have regarding the methodology proposed by the Authors and would appreciate some additional information in the text regarding these points before I could recommend publication.*

We thank referee#1 for this overall positive comment and all other comments which helped to improve the manuscript. For the revised manuscript we checked the English language and clarified several issues (see below for our specific comments). Currently, we perform a final proofreading before uploading the revised manuscript.

*1. As noted by the Authors in their introduction, sensitivity analysis and tagging approach are two approaches that are used to answer different questions. Sensitivity deliver impacts whereas tagging delivers contributions. While it is rather clear that impacts can be used to inform on the potential effects of emission reductions on air quality levels, it is rather unclear how the contributions estimated for the Authors can be used in practice. In one of their earlier work, the Authors mentioned the possibility of using contributions in complement to the impacts to inform on the potential of emission reductions that go beyond the threshold covered by the perturbation or sensitivity method. But this possibility is not mentioned in this work. On the contrary, confusion is introduced in some sections in which the Author seem to indicate that contributions can be used to support air quality strategies, e.g. in Section 4.2 (first three lines).*

Reply: Contribution analyses provide no direct information about potential benefits from emission reductions (see also Thunis et al., 2019). As discussed by Mertens et al. (2018), which was mentioned by referee#1, the combination of the sensitivity approach with the tagging approach can help to better understand the changes in atmospheric composition by specific emission reductions. The goal of this manuscript, however, is not to investigate potential mitigation options. The goal is to quantify the contribution of the (current state of) land transport emissions to ozone and ozone precursors. The tagging method is well suited to answer this question. Such a quantification of the current status is, to our understanding, the first step in understanding the influence of different emission sources on the atmospheric composition, but can of course not replace

additional sensitivity simulations. We clarified this in the revised manuscript.

The new section in the Introduction reads:
**In contrast to this, Dahlmann et al. (2011) and Mertens et al. (2018) have used a source apportionment method (by a tagged tracer approach, called tagging hereafter) to calculate the contribution of land transport emissions to ozone. The perturbation approach is based on a Taylor approximation to estimate the sensitivity of ozone (or other chemical species) at a base state (w.r.t. the chemical regime) to an emission change. The tagging approach, however, attributes all emissions at any base state (w.r.t. the chemical regime) to the corresponding tagged emissions, but gives no information about the sensitivity of ozone to an emission change (see also, Grewe et al., 2010). For a chemical specie that is controlled by linear processes, the perturbation and the tagging approaches lead to identical results, however, the ozone chemistry is strongly non-linear. Therefore, only for small perturbations around the base state (w.r.t. the chemical regime) the response of ozone on a small emission change can be considered as almost linear, but the perturbation approach does not allow for a complete ozone source apportionment (e.g. Wild et al., 2012). As an example, Emmons et al. (2012) have reported that tagged ozone is 2–4 times larger than the contribution calculated by the perturbation approach. As has been outlined in numerous publications, this difference is due to different questions these methods answer. The perturbation approach investigates the impact of an emission change on the mixing ratios of ozone and is therefore well suited to evaluate for example mitigation options. The tagging approach quantifies the contribution of specific emission sources onto the ozone budget for a given state of the atmosphere (Wang et al., 2009; Emmons et al., 2012; Grewe et al., 2017; Clappier et al., 2017; Mertens et al., 2018). These contributions do, however, not necessarily change linearly with potential changes in emissions. The difference between the results from the perturbation and tagging approaches can actually be used as an indicator for the degree of non-linearity of the chemistry as pointed out by Mertens et al. (2018) in their equation 6. In the following we use the terms 'impact' to indicate results from perturbation approaches and 'contribution' to refer to results of tagging methods. In this study, we are interested in the contribution of land transport emissions to ozone in Europe. Therefore, we chose a tagging method for source apportionment.**

In addition we clarified this at several parts of the manuscript (see 'diff' version). Especially the first part of Section 4.2 reads now: "**To improve the understanding of extreme ozone events, ...**"

*2. Along the same lines, the Authors mention that these findings based on*

*tagging are in line with other studies using perturbation methods (see l27 in the discussion section). How can these conclusions be reached when it is clearly mentioned in the introduction that perturbation methods and tagging are expected deliver different results. These two statements contradict each other, unless O3 may be considered as a linear species, in which case both methods would indeed converge to the same conclusions*

Reply: We agree with referee#1 that this part of the conclusions was missleading in the original manuscript and your comment is very much in line with the comment from referee#2. We rewrote large parts of the conclusions including a more detailed comparison of the results from different tagging methods for Europe. This comparison helped to understand the results of the different tagging approaches in more detail. The new part of the discussion reads:

[revised manuscript text omitted]

*3. In some sections, many numbers are given to characterize the various contributions, e.g. Section 5. A few additional lines to detail the implication these results may have would be useful.*

Reply: We thank referee#1 for this suggestion. Generally, our manuscript is composed in such a way that the first sections present only our findings while we discuss the implications in the following sections. Concerning Section 5 we added the following sentence:

**These results indicate the importance of land transport emissions for the mixing ratios of reactive nitrogen levels in German cities. Further, they clearly show the importance of fine resolved emission inventories (and models) for source apportionment of short lived chemical species.**

*Most of these comments address spelling errors or unclear grammatical sentences. But I would strongly suggest the Authors to review the whole text regarding the English writing.*

Reply: Thank you very much for the corrections. We added all of them. For the revised manuscript we checked the manuscript carefully and will perform a final proofreading before uploading it. We are very sorry for the large amount of spelling errors in the original manuscript.

*1. In many sentences 'as' is used in place of 'than' (e.g. p19 l27; p22 l6; p26 l25. . .)*
Reply: We checked the whole manuscript and fixed it (hopefully) everywhere.

*2. P1 l28: teh - the*
Reply: Fixed !

*3. P3 l20: quantifies - quantified* Reply: Fixed !

*4. P4 l29: Th - the*
Reply: Changed to 'to'

*5. P5 l34: an - a*
Reply: Fixed

*6. P5 l35: to - too*
Reply: Fixed

*7. P8 l9: not - note*
Reply: Fixed

*8. P8 l8: party - part*
Reply: Changed to parts

*9. P19 l16: kept - be kept*
Reply: Fixed

*10. P19 l 25: the text within parentheses is unclear*
Reply: Changed to **(i.e. the emission sectors anthropogenic non-traffic and aviation)**

*11. P19 l29: increase - increases*
Reply: Fixed

*12. P19 l34: all most - almost*
Reply: Fixed

*13. P22 l14 & 15: sentence is unclear*
Reply: We changed the sentence to: **We analyse the contribution of land transport emissions to the ozone budget in Europe by investigating the net ozone production, which is defined as:**

*14. P23 l3: second - second most*
Reply: Fixed

*15. P23 l11: is displayed - are displayed*
Reply: Fixed

*16. Discussion section: could the Authors add a few words to explain how all these contribution numbers can be validated? Can we use contributions to know which inventory might be closer to the truth?*
Reply: This is indeed a good point. What can be done is an evaluation of model results and diagnosed contributions with measurements for specific periods to check, if processes are implemented correctly (or if they are missing). Examples could be periods with large influence of stratospheric ozone (where models should show large stratospheric contributions) or measurements in city plumes, for which models should show a large contribution of ozone from anthropogenic categories. We added a short discussion on this in the revised manuscript. A crucial point is also the differences in the tagging methods, which need to be investigated in more detail to understand strengths and weaknesses of the different approaches better. In our opinion, contributions alone do not help to discuss individual emission inventories. At the end all information (measured and simulated ozone mixing ratios, and contributions) can help to estimate if emission inventories are in a plausible range. However, in our opinion they cannot help to judge, if an emission inventory is right. The additional part in the discussion reads:

**Challenging remains also the question on how to evaluate these source apportionment results. Clearly, a comparison of different source apportionment methods would help in revealing individual strengths and weaknesses of the methods. In addition, we plan to include source apportionment results in the process of model evaluation (and suggest similar to other modelling groups). By comparing measurements and model results for specific episodes or for specific regions (e.g. in plumes of cities, in regions with strong lightning activity or events of stratospheric intrusions) it can be investigated, if the diagnosed contributions are in a plausible range. Further, the influence of model biases on the analysed contributions can be estimated. A direct evaluation of these contributions, however, is not possible.**

*17. P25 l3: corresponds - correspond*
Reply: Fixed

*18. P25 l7: depend - depends*
Reply: Fixed

*19. P25 l8: contributor - contributors*
Reply: Fixed

*20. P25 l17: increase - increases*

Reply: Fixed

*21. P25 l19: regions of - regions with*
Reply: Fixed

*22. P25 l25: not largest - not the largest*
Reply: Fixed

*23. P27 l11: by different - between different*
Reply: Fixed

*24. P27: l28 to 30: please check the use of the word 'uncertainty' which is used many times in a couple of sentences*
Reply: We rephrased the sentences to:
**Of course, also the uncertainties in the emission inventories for emissions outside of Europe can influence the contribution analyses considerably, but this has not been investigated in the present study. During summer the differences between the contributions diagnosed using the two emission inventories are larger than the year-to-year variability. Hence, during summer uncertainties of emission inventories for Europe influence the contribution analyses considerably.**

*25. P28 l2: studies - studied*
Reply: Fixed

*26. P28 l9: o - ?*
Reply: Fixed

*27. P28 l9 region - regions*
Reply: Fixed

*28. P28 l11 increase - increases*
Reply: Fixed

*29. P28 l19-20: can the Author develop a little bit more on how they plan to use observation data to validate the contributions? I believe this is a key point and one of the major benefits of the tagging approach.*
Reply: For the revised manuscript we added some more details about this in the discussion section (see our reply above). In the conclusion section we added a reference to the discussion section.

We are looking forward to your reply,
Mariano Mertens
(on behalf of all co-authors)

*Mertens et al. perform a source attribution study examining the contribution of different emission sectors to air pollution over Europe, with a focus on ozone as a pollutant, a special focus on emissions from the road transport sector, and a regional focus on Europe and Germany. They employ a uniform methodology for "tagging" the emissions of ozone precursors in a system of coupled models, allowing a consistent downscaling to be made from the global scale to the national scale. Furthermore, they compare two simulations performed with different emission inventories, showing the sensitivity of the sectoral contributions to the way in which the emissions from each sector are represented in the emission inventory. This combination of sensitivity and source attribution reveals some interesting information about the behaviour of of tropospheric ozone in the model system used, for example the particularly strong differences in the contribution of land transport emissions to the higher percentiles of the ozone distribution when a spatially more explicit inventory is used.*

Reply: We thank referee#2 for this elaborate summary and the detailed review which helped to improve the manuscript considerably.

*The manuscript is clearly within the scope of ACP, and the method clearly has a lot of potential to inform international air quality policy. Unfortunately the manuscript in its current form suffers from a number of serious flaws, which must be corrected before it can be accepted for publication.*

Reply: We thank you for your overall positive comment. In accordance with your comments (see below for details) and the comments from referee#1 we strongly revised parts of the manuscript. Currently, a final proofreading is performed after which we will upload the revised manuscript. We hope it can then be accepted for publication.

*Firstly, the quality of the written English is terrible. The manuscript is littered with grammatical and spelling errors, and written in a generally inaccessible style. I do not feel that it is my job as a reviewer to provide an exhaustive list of these errors. The authors should seek additional help to get the language up to an acceptable standard. I will give one example though: the very title of the manuscript contains a jarring error. The current title basically implies that ozone causes land transport emissions. Clearly this is the other way around. Land transport emissions happen first, and this leads to ozone production. A grammatically correct title could be "Attributing ozone and its precursors to land transport emissions in Europe and Germany".*

Reply: We are really sorry for the errors in the first draft of the manuscript. Of course it is not the work of the referees to perform a language editing. We checked the manuscript in detail and corrected many errors. In addition we revised the title according to the suggestion of referee#2.

*In the abstract, the authors state that tagging is "required" and that their method is the "only possible" way to examine global to regional scale effects. This language is way too strong and should be toned down before publication. This is especially true given that the authors themselves state on line 28 of page 25 that their results are "consistent" with a perturbation study, and also given the fact that the experiment design doesn't actually make a distinction between land transport emissions in Europe and the rest of the world.*

Reply: We think that the scientific community agrees that tagging methods are the only correct way to calculate contributions (for non-linear species). Impacts and contributions can be similar, but they answer completely different questions. Therefore we don't agree that the language of the sentence '[...]the contribution of land transport emissions to tropospheric ozone cannot be calculated or measured directly, instead atmospheric-chemistry models equipped with specific source apportionment methods (called tagging) are required' is too strong. Yet, we rephrased the other sentence to:

**We investigate the combined effect of long range transported ozone and ozone which is produced by European emissions by applying the tagging diagnostic simultaneously and consistently on the global and regional scale.**

*Page 2, lines 26-27: while this is generally true on very small scales (eg. urban areas), the response of ozone to perturbation of precursor emissions in remote regions has been shown to be approximately linear. See for example Wild et al. (2012) and Turnock et al. (2018). Since the authors are also discussing long-range transport, some additional discussion of this here would be relevant.*

Reply: We agree with referee#2 that in remote regions (i.e. with low $NO_x$ mixing ratios) the ozone chemistry is (almost) linear w.r.t. $NO_x$ and VOC perturbations. However, with increasing $NO_x$ mixing ratios the chemistry of ozone cannot be considered any more as linear (see for example Fig. 1 in Grewe et al., 2012). Concerning your comment we think it is essential to further discuss the differences of the tagging and the perturbation method to clarify this point. The perturbation approach is based on a Taylor approximation around a base state w.r.t. the chemical regime (called $x_0$). The goal of this approach is to estimate a sensitivity (e.g. $dO_3/dE$, where E are the emissions) of the ozone chemistry around $x_0$ by a Taylor approximation. This sensitivity can be used to estimate a response of ozone on emission changes (as done by Wild et al., 2012). Clearly, this approximation is only valid around $x_0$, but not for a different base state $\tilde{x}_0$. Further, only for small perturbations non-linear effects can

be neglected (first elements of Taylor series). This means that in regions with large $NO_x$ emissions non-linear effects can be neglected only for very small perturbations (e.g. 5 %) around $x_0$. As the approximation is only valid around $x_0$ an extrapolation to larger perturbations leads to larger errors. Therefore, Wild et al., 2012 introduces a non-linearity factor (equation 6 therein) to account for non-linearities for much larger perturbations as the original 20 %. Further they state: 'For emission reductions greater than 60 % this correction remains insufficient, and we do not expect the parameterization to work as well under these conditions.' Clearly, for such large perturbations the Taylor approximation is not valid anymore. This has been discussed in great detail also by Grewe et al., 2010. The tagging approach, however, works in a completely different way. It does not consider the sensitivity of the ozone chemistry to an emission change. Instead, it attributes ozone at any base state w.r.t. the chemistry $x_y$ to the corresponding emissions. Thus, the non-linearities are implicitly taken into account. However, the tagging approach gives no information about the sensitivity of an emission change (e.g. $dO_3/dE$). In addition, Wild et al. (2012) clearly state:

'The 20 % emission perturbations applied in the HTAP studies were chosen to be small enough to give an approximately linear response while being sufficiently large to provide robust signals in all models. However, the response of O3 to its precursor emissions is known to be non-linear (e.g. Lin et al., 1988), and it is important to characterize where these non-linearities become significant. Scaling a 20 % emission reduction by a factor of five has been shown to underestimate the response to a 100 % reduction (Wu et al., 2009), and while this underestimation is relatively small for VOC emissions, generally less than 10 %, it can exceed a factor of two for NOx emissions (Wu et al., 2009; Grewe et al., 2010), and shows a strong seasonal dependence (Wu et al., 2009). For this reason the sensitivity approach used in the HTAP studies is unsuitable for deriving a full source apportionment for $O_3$. However, it does not preclude its use in estimating the impact of less severe emission changes.'

This is clearly in line with our argumentation. Yet, to clarify this point in the revised manuscript we changed the paragraph accordingly:
**For a chemical specie that is controlled by linear processes, the perturbation and the tagging approaches lead to identical results, however, the ozone chemistry is strongly non-linear. Therefore, only for small perturbations around the base state (w.r.t. the chemical regime) the response of ozone on a small emission change can be considered as almost linear, but the perturbation approach does not allow for a complete ozone source apportionment (e.g. Wild et al., 2012). As an example, Emmons et al. (2012) have reported that tagged ozone is 2–4 times larger than the contribution calculated by the perturbation approach. As has been outlined in numerous publications, this difference is due to different questions these methods answer. The perturbation approach investigates the impact of an emission change**

on the mixing ratios of ozone and is therefore well suited to evaluate for example mitigation options. The tagging approach quantifies the contribution of specific emission sources onto the ozone budget for a given state of the atmosphere (Wang et al., 2009; Emmons et al., 2012; Grewe et al., 2017; Clappier et al., 2017; Mertens et al., 2018). These contributions do, however, not necessarily change linearly with potential changes in emissions. The difference between the results from the perturbation and tagging approaches can actually be used as an indicator for the degree of non-linearity of the chemistry as pointed out by Mertens et al. (2018) in their equation 6. In the following we use the terms 'impact' to indicate results from perturbation approaches and 'contribution' to refer to results of tagging methods. In this study, we are interested in the contribution of land transport emissions to ozone in Europe. Therefore, we chose a tagging method for source apportionment.

*Page 4, lines 3-4: Aren't the last two points in this list in fact exactly the same thing?*

Reply: We are sorry for the confusion the sentence caused in the original manuscript. Point 3 is dedicated to year to year variability (e.g. years with large biomass burning emissions or summer heatwaves). Point 4 is dedicated to the seasonal variability (e.g. strong biogenic emissions in summer). To clarify this we rephrased the part in the revised manuscript (see our reply to the next point).

*Page 4, line 5: I can see how using two different inventories can somewhat account for uncertainties in the emissions, but three years is way to short a period to account for interannual variability. I also do not see how the model uncertainty or the uncertainty in the choice of source apportionment method is accounted for at all in this experiment design. It's fine to mention that there can be a lot of uncertainty, but the authors should not claim to be doing more to address these uncertainties than they actually are.*

Reply: Of course three years are not enough to catch the full range of interannual variability. Referee#2 is completely right that we do not account for model and/or methodological uncertainties (e.g. different source apportionment methods). During the writing process of the manuscript we changed the order of the four points, but forgot to change the sentence on p4 l5. We clarified this accordingly. The changed paragraph reads:

Typically, the uncertainties of such source apportionment studies are large. Reasons are:

- uncertainties in the models (e.g. chemical/physical parametrizations) and trough the choice of source apportionment methods;

- uncertainties of the emissions inventories;

- seasonal variability of the contributions caused by meteorological conditions and seasonal cycles of emissions (e.g. stronger biogenic emissions and more active photochemistry during summer than winter);

- year to year variability of the contributions caused by meteorological conditions or large emissions of specific sources in specific years (for example yearly differences of biomass burning emissions);

To account for the uncertainties due to different emission inventories we performed simulations with two different anthropogenic emission inventories. To further account for the seasonal variability we investigate the contributions for winter and summer seasons. In addition, we consider always three simulation years to gain insights in the variability of the contribution in different years. The investigation of uncertainties caused by models and/or source apportionment methods is beyond the scope of this study.

*Page 5, lines 32-35: These are the only lines in the paper where the authors discuss model evaluation. I understand that the model has been evaluated elsewhere, and the model is basically as good/bad as other models, but I would appreciate some more discussion about how the model performance could be expected to influence the conclusions of the manuscript. Since the authors also want to use their model to examine extreme ozone events (in Section 4.2), there must be at least some analysis of how well the model is capable of representing these events in comparison with observations.*

Reply: We added section (Sect. 2.5) including a short evaluation of simulated ozone concentrations in comparison to Airbase data. This Section reads:

[revised manuscript text omitted]

*authors, and how this could be expected to influence the results of the study is required already in Section 2.1. Similarly, since the authors are also considering the global scale, they should also put their methodology into the context of the existing techniques for source attribution at the global scale. The authors already cite Emmons et al. (2012) elsewhere in the paper, but do not mention this work in Section 2.1, where it would be appropriate to have some discussion of how these methods differ, and how this might influence the results of the study. One very important difference is that Emmons et al. (2012) only consider NOx as a precursor of ozone, while the technique employed by the authors combines the effects of both NOx and VOC precursors. Similarly, the study of Butler et al. (2018) is also missing from the discussion. Butler et al. (2018) account for effects of both NOx and VOC as ozone precursors, but they make some very different design decisions to the technique employed by the authors. The authors must do more to put their method in the context of the previous work, and discuss the relative strengths and weaknesses of the approach they have chosen.*

Reply: The method we apply has been discussed in detail by Grewe et al. (2017). Therefore, the intention of this section was only to recap the general idea of the applied tagging method to the reader. It was never indented as a full discussion of our tagging approach compared to other approaches. However, we agree with referee#2 that a short discussion about the different approaches is helpful here, as these differences will also be discussed in the discussion. The newly added part reads:

**Some of the categories listed in Table 3 are not directly associated with emission sectors. These categories are stratosphere, $CH_4$ and $N_2O$. All ozone which is formed by the photolysis of oxygen, i.e**

$$O_2 + hv \longrightarrow O(^3P) + O(^3P), \tag{1}$$

is labelled as stratospheric ozone. The degradation of $N_2O$ is a source for $NO_y$ (and loss a of ozone) by the reaction:

$$N_2O + O^1D \longrightarrow 2NO. \tag{2}$$

The degradation of $CH_4$ is considered as source of $NMHC^{CH_4}$. This refers to the reaction:

$$CH_4 + OH \longrightarrow CH_3O_2 + H_2O. \tag{3}$$

As have been discussed recently in detail by Butler et al. (2018) all tagging methods are based on specific assumptions and have specific limitations. The scheme of Grewe et al. (2017), which we apply in the current study, is based on specific assumptions, which differ from other tagging schemes used in regional and global models. One important difference is the question whether ozone formation is attributed to $NO_x$ or VOC precursors. The schemes which are available in the regional models CMAQ (called CMAQ-ISM, Kwok et al., 2015) and CAMx (called CAMx OSAT, Dunker et al., 2002) use threshold conditions to check, whether ozone formation is $NO_x$ or VOC limited. Depending on this the production is attributed to $NO_x$ or VOC precursors only. The scheme of Emmons et al. (2012), applied on the global scale, tags only $NO_x$ and therefore ozone production is only attributed to $NO_x$ precursors. Based on the work of Emmons et al. (2012), Butler et al. (2018) presents a scheme, which attributes ozone formation either to $NO_x$ or VOCs (implying that usually 2 simulations, one with $NO_x$ and one with VOC tagging, are performed). This scheme has also been applied by Lupaşcu and Butler (2019) in a regional model simulation over Europe, using only the $NO_x$ tagging scheme. Compared to these schemes the scheme of Grewe et al. (2017) attributes ozone production always to all associated precursors (i.e. NOx, $HO_2$ and VOCs) without any threshold conditions. In VOC limited regions, this approach leads to the effect that a $NO_x$ emission reduction of an emission sector reduces the contribution of that sector, and increases the contribution of the other sectors. In contrast, a reduction of VOC emissions decreases the contribution of the respective sector only. The latter is similar to the approaches integrated in CMAQ or CAMx, which attribute ozone production in the case of a VOC limit to VOC precursors only. Compared to a $NO_x$ tagging, our approach leads to lower contributions of $NO_x$ sources, since they compete, not only with other $NO_x$ sources, but also with VOC sources. Because of the family concept, which is necessary to keep the memory consumption and the computational costs low, the tagging method applied in our study can lead to some unphysical artefacts. As an example, Grewe et al. (2017) discuss the production of PAN by NMHCs from $CH_4$ degradation. Further, due to the combinatorial approach for instance also NMHCs from stratospheric origin can occur in small

amounts, which is also an unphysical artefact. The main reason for this is the definition of the PAN family, which transfers tags from $NO_y$ to NMHCs. Other tagging schemes have specific issues as well. As an example, the scheme of Emmons et al. (2012) does not neglect the $O_3$-$NO_x$ null cycle, which leads to an overestimation of local sources compared to long range transport sources (see also Kwok et al., 2015). Overall, the impacts of the underlying assumptions on the results are difficult to quantify. Therefore, it is important to study effects of different emission sources with different methods (at best in the same model framework), in order to understand better the strengths and weaknesses of the different approaches and their impact on the source apportionment results.

*Also in Section 2.1, the authors could briefly mention how stratospheric ozone is tagged in their approach, since this does not fit into the framework of their Equation 2.*

Reply: We added a note about tagging of stratospheric ozone in the revised manuscript (see reply above).

*Section 2.2: The authors should make it clear that the tags are applied globally, with no distinction between emissions in Europe and the rest of the world. This is acknowledged later in the manuscript, but the reader would benefit from having this made clear already in this section.*

Reply: We added a note in the Section which reads:

In the configuration of the tagging method applied for the present study we use only one global tag for every source category. While this allows to investigate the contributions of all global emissions of a specific emission source to ozone mixing ratios, we are not able to separate contributions from local and long range transport (i.e we cannot separate contributions from, for example, European and Asian land transport emissions to European ozone levels, but we can quantify the contribution of global land transport emissions to European ozone levels).

*Page 10, lines 17-24: For some additional context here, it would be nice to know how the proportional contributions of land transport to ambient modelled NOy compare to the proportional contribution of land transport to total NOx in the inventories. Is the contribution as would be expected from simply looking at the emissions, or is it disproportionally higher or lower?*

Reply: This is indeed a good question. We calculated the relative share of land transport emissions to all anthropogenic + soil $NO_x$ emissions for June (see Fig. S2, which we add also to the revised Supplement). The contributions of

[Figure]

Figure S2: Relative contribution of land transport $NO_x$ emissions to all other emissions (considering soil-$NO_x$, shipping, anthropogenic, AWB and biomass burning emissions; in %) for July 2009; (a) for the MACCity emission inventory and (b) for the VEU emission inventory.

land transport emissions are in the range of 50 % to 70 %. The contribution is larger in the VEU emission inventory compared to the MACCity emission inventory. The contributions of the emissions are in a similar range as the contribution of land transport emissions to $NO_y$, however, the regional distribution differs slightly. Near the hot-spots (e.g. Paris) we found smaller relative contributions of land transport emissions to $NO_y$, while the values are larger in rural areas. We added a note about this in the revised manuscript:

**The relative contribution of land transport emissions to ground level $NO_y$ is in the range of 40 % to 70 % in most parts of Europe (see Fig 4). These relative contributions are similar as the share of land transport $NO_x$ emissions to all $NO_x$ emissions (see Fig. S9 in the Supplement), but compared to the share of the emissions the contributions to $NO_y$ are slightly lower near hot-spots, and larger in rural areas.**

*Section 4.1, page 15: The authors rightly interpret the ozone due to land transport in DJF as coming from long-range transport. I also understand that the limits of the experimental design (one global tag for land transport) make it hard to say anything about long-range transport in JJA, when local photochemistry is more active. But could it be possible to try? For example, could they look at the land transport contribution at the western boundary of the refined grid in JJA, and use this as a rough estimate of the contribution of land transport (and other sectors) in remote regions to baseline ozone in Europe? This could add a lot of value to the study and would be highly relevant for international policymaking.*

Reply: This is indeed a good point. As mentioned by referee#2 we cannot directly estimate the relative importance of 'global land transport emissions' compared to 'European land transport emissions', as we consider only one global tag. To answer this question in detail more tags would be necessary. Based on your suggestion we added a Figure to this Section (see Fig S3) in which we show area averaged contributions for different categories and for different regions. One region we defined here is called inflow and spreads over a large area of the western boundary of the finer domain. We added a paragraph describing this in the manuscript. As you see, we added also a separation between Northern Alps and the Po Basin (see our answer below).

**To quantify the contributions of land transport emissions and other emission sources in different regions in more detail, Fig. S3 shows area-averaged relative contributions for JJA and DJF for the *REF* and *EVEU* simulations (absolute contributions are given in Table S1 to Table S8 in the Supplement). The geographical regions were defined according to the definitions of the PRUDENCE project (Christensen et al., 2007), but slightly modified. The region Alps was split up in two separate regions called 'Northern Alps', defined as rectangular box ($46° : 48°$ N and $9° : 13°$ E), and 'Po Valley' ($44° : 46°$ N and $5 : 15°$ E). Note, however, that the region Northern Alps contains parts of Switzerland and Southern Germany, which are still rather flat and subject to large land transport emissions. In addition, we defined a region called 'inflow' ( $40° : 60°$ N and $-13° : -11°$ E). This region is used to quantify contributions in the air advected towards Europe. A figure summarizing the definition of all regions is part of the Supplement (Fig. S12). The relative contribution of land transport emissions in the 'inflow' region is about $9$ % and very similar in both seasons and for both European emission inventories. During DJF the contributions in all regions are very similar. During JJA the contribution of land transport emissions increases in most regions compared to the 'inflow' ($\approx 9$ %). In the Po Valley the contribution reaches up to 16 %. Unfortunately, the difference between the contribution in a specific region compared to the contribution in the region 'inflow' cannot be used to calculate $O_3^{tra}$ from European emissions. Such a calculation requires different tags for global and European land transport emissions. The relative contribution of other anthropogenic emissions in the 'inflow' region ($\approx 34$ %) is also very similar in both seasons. During DJF the contributions in the different regions remain very similar to the contributions in the 'inflow' region. During summer, in contrast, a West-East gradient of the contribution of anthropogenic emissions is present over Europe with a decrease of the contribution of up to $\approx 27$ % in Eastern Europe. This decrease is mainly caused by the seasonality of the different emissions (discussed further below). The biogenic emission category shows different relative contributions in the 'inflow' region during DJF ($\approx 11$ %) compared to JJA ($\approx 14$ %), which is mainly**

[Figure]

Figure S3: Relative contributions to ground-level ozone (in percent) area averaged in different geographical regions for DJF 2008 to 2010 (triangles) and JJA 2008 to 2010 (squares). Shown are the results of the *REF* (blue) and the *EVEU* simulations (red) for (a) the land transport category, (b) the anthropogenic emissions, (c) the biogenic category, and (d) all other categories. For simplicity the anthropogenic contains the categories anth. non-traffic, aviation and shipping. The residual contains all other categories. The vertical-axis scale differs for (a) to (d).

**caused by the strong increase of biogenic emissions during summer compared to winter. In the different regions the relative contributions increase during JJA compared to DJF, and, compared to the 'inflow' up to $\approx 20$ %. The contribution of all other tagging categories during DJF is around $\approx 47$ % in most regions, and ranges between 41 % and 36 % during JJA.**

*Page 16, line 3: the seasonal cycle of photochemical activity also plays a role here.*

Reply: Indeed. We changed this sentence to:

This seasonal cycle is caused by a complex interplay of the seasonal cycles of different **emission sources, meteorology and photochemical activity.**

*Page 16, line 8: is there any way in this study to separate the influence of soil NOx and biogenic VOC? Or are these two different sources inextricably joined together into the "biogenic" sector?*

Reply: As they are emitted in the same category there is no possibility to separate them anymore from the simulation results. However, we see where this can deliver important insights, and we are currently revising the tagging method in such a way that these two emissions could be handled separately.

*Section 4.2: As mentioned earlier, it would be nice to know how well the model is capable of reproducing the extreme values of ozone as measured. If the model is doing a good job at this, then the results reported here could help to understand these extreme ozone measurements. If the model is not doing well at this, then the results reported here could potentially provide information about systematic model biases, and point the way towards improving the model. As it currently stands, it is not clear at all how these results should be interpreted.*

Reply: As discussed above we added a short section with a model evaluation to the revised manuscript. This evaluation shows that the model is able to reproduce the measured 95th percentile of ozone values quite well on the rural scale, but for strong local ozone enhancements the resolution of our model is too coarse (e.g. Tie et al., 2010). The evaluation clearly shows that the model results are not well suited for analysis of the contributions during extreme ozone events on the levels of individual cities. In our analysis, however, we focused on larger geographical regions. We think that on the basis of these larger geographical regions the model results are well suited to investigate the general trends of ozone contributions. Further, our finding that the relative contribution of land transport emissions increase during extreme ozone events compared to the mean conditions is in line of Valverde et al. (2016). They reported a large importance of land transport emissions during high ozone events for Barcelona and Madrid surroundings.

*Page 19, line 11: The region "Alps" includes the Po Valley. Does this mean that high mountains are in the same region as a polluted valley? The influences on air quality would be expected to be very different in these regions. High mountains will be more influenced by the free troposphere (and long-range transport), while the valley will be more influenced by local sources. Furthermore, "Alps" and "Po Valley" are used individually in this section and elsewhere in the manuscript. It is not always clear which region is meant. The authors could consider disaggregating this region into two sub- regions for their analysis (which could be quite informative), or at least being clearer about exactly which region they are referring to throughout the text.*

Reply: Yes indeed, the region called 'Alps' includes the Po Valley and the Alps. The main intention for this was that we wanted to stick to the geographical

regions defined in the PRUDENCE project. However, we agree that from the point of view of air quality these regions strongly differ. To take this into account, we split the region 'Alps' in two subregions called 'Northern Alps' (defined as $44° : 48°$ N and $5° : 15°$ E) and 'Po Valley' (defined as $44° : 46°$ N and $9° : 13°E$) . However, the results for both regions are still very similar. The main reasons are:

- The region Northern Alps contains parts of Southern Germany and also Switzerland, were the mountains are not very high, and much traffic is present.

- Even in the 'higher alps' there are some very important roads with large land transport emissions (e.g. Brenner and Inn valley) which can be clearly seen in the emission inventory. On the 50 km resolution these emissions are mixed over quite large regions.

To better represent the sharp contrast between Alps and the Po Valley a much finer resolution (and fine resolved emission inventories) are necessary, which pose challenging tasks for the future.

*Page 19, lines 15-16: the discussion about "uncertainties" in the inventory is very vague here. Could the large range in the contribution of land transport to extreme ozone when using EVEU emissions be related to the higher spatial heterogeneity and existence of more "hot spots" in this inventory compared with REF? There could potentially be some important information here about the need to get the distribution of NOx right in order to capture the high ozone events. A comparison of the REF and EVEU ozone timeseries with some measurements from urban background stations during extreme events could potentially add a lot of value here.*

Reply: It is indeed interesting to investigate how different geographical distributions of $NO_x$ emissions could influence the ability of the model to simulate high ozone events. This issue has partly been investigated in previous publications (Tie et al., 2010; Markakis et al., 2015). Compared to these previous studies, the resolution applied here is rather coarse (50 km). The 95th percentiles of ozone for *REF* and *EVEU* are rather similar (see also the newly added evaluation section). When comparing individual stations during specific periods, we noticed that maximum ozone values are not better represented by EVEU compared to REF.

As the 95th percentiles of the ozone values are very similar, we think that the differences of the contributions between the two emission inventories are only caused by the different geographical and sectoral distributions. To clarify this we rephrased and extended the discussion. However, for follow up work we agree that this is still an interesting question and should be further investigated using an improved model set-up at finer resolution. The modified text reads:

The ozone values at the 95th percentile (see Sect. 2.3) and at the other percentiles (see Figs. S1 and S2 in the Supplement), however, are similar for *REF* and *EVEU* (i.e. none of the emission inventories leads to strongly different representation of extreme ozone events in the model). Accordingly, the discussed differences of the relative contributions are not caused by a different representation of the ozone values themselves, but only due to the different geographical and sectoral distributions of the emissions in *REF* and *EVEU*. This demonstrates the large uncertainty, especially for contributions during high ozone events, of the source apportionment analyses which is caused by the uncertainties of emissions inventories (e.g. geographical distribution of emissions, total emissions per sector). These uncertainties must be taken into account in source attribution studies focusing on high ozone events.

*Page 23, line 4: the results are not "rather similar", but actually have some important differences, which are subsequently discussed. I think what the authors are trying to say here is that the contribution of land transport is similar in each case, but this is not the meaning which comes across.*

Reply: We agree with referee#2 that our original intention of the discussion does not come across. Also the comment from referee#1 shows that this part of the discussion caused confusion. Therefore we revised this part of the discussion completely, taking also into account some more (recently published) work to discuss potential reasons for the differences between the results of the different source apportionment results (taking also into account one of the next comments from referee#2). The new part of the discussion reads:

[revised manuscript text omitted]

*Page 25, lines 25-26: This sentence basically conveys no meaning and could be easily deleted with no loss to the manuscript. Alternatively the authors could try to be clearer about what they mean here.*

Reply: We rephrased the sentence to:

The result that regions are hot-spots for $NO_y$ from land transport emissions, but not for $O_3$ from land transport is counter intuitive. The reasons for this is that large amounts of $NO_x$ emissions alone

**are not sufficient for large ozone production. This is caused by the non-linearitiy of the ozone chemistry and the strong interdependence of ozone production and meteorological conditions (e.g. Monks et al., 2015).**

*Page 25, last paragraph: if the previous work only accounts for the contribution of Euro- pean land transport emissions to European ozone, and the current study also includes global emissions, then shouldn't the current study result in a higher contribution than the previous work? The opposite appears to be the case. Can the authors explain this apparent discrepancy?*

Reply: We agree. As already discussed above, we tried to clarify this in the new discussion. The differences between the studies, however, are so large that we cannot fully explain the discrepancies, but the discussion hopefully provides some insights. What is really needed to understand the differences between the tagging methods is a detailed inter-comparison of them.

*Page 26, line 23: the authors appear to be concluding from the strong influence of the "biogenic" sector that soil NOx emissions are strongly influencing ozone. But couldn't this also be biogenic VOC? How do they separate the influence of these two different sources? A comparison with Butler et al. (2018) could be instructive here, since in that study the separate roles of NOx and VOC as ozone precursors were examined. Comparison of their Figure 3 and Figure 4 indicates that biogenic VOC make a larger contribution to European ozone in summer than biogenic NOx. The authors should discuss this here.*

Reply: Of course also biogenic VOCs are very important for the ozone production. As discussed above, we cannot differentiate between ozone produced by biogenic VOCs and soil-$NO_x$, as we join them together in one category. The soil-$NO_x$ emissions are an important contributor to the $NO_x$ emissions in Europe in summer (see Fig. S4 in the Supplement). Uncertainties of these emissions cause uncertainties of the simulated contributions. However, also the biogenic VOC emissions are uncertain. Therefore, we rephrased this part to clarify that we do not want to say that biogenic VOCs are not important. The discussion of the importance of biogenic VOCs and $NO_x$ with the reference to Butler et al. (2018) was already introduced in the revised discussion above.

*Page 28, lines 14-15: the future work proposed by the authors would indeed be ex- tremely interesting from a policymaking perspective. If possible, they should also in- clude as many other sectors as possible. This could help to inform decisions about where emission reductions would be most effective.*

Reply: Thanks for this positive comment. Actually, the main focus is on land transport, but we intent to investigate also other categories.

*Page 28, lines 28-29: again, it appears that the authors are over-interpreting*

*their results when they conclude that soil NOx has a strong influence on European ozone levels.*

Reply: We guess referee#2 meant lines 18-19. Here we write: 'Here, the focus should not only be on the land transport emissions, but also on other important emissions, including especially biogenic and soil-$NO_x$ emissions, which have large uncertainties and contribute strongly to European ozone levels.' We do not see where we over-interprete the results. However, to make the sentence more clear we add 'VOC' to biogenic:

'Here, the focus should not only be on the land transport emissions, but also on other important emissions, including especially biogenic **VOCs** and soil-$NO_x$ emissions, which **are subject to** large uncertainties and contribute strongly to European ozone levels. '

We are looking forward to your reply,
Mariano Mertens
(on behalf of all co-authors)

**References**

[revised manuscript text omitted]

---

## Author Response (AR2)

Dear Andreas Hofzumahaus

Thank you very much for guiding through the editorial process.

To clarify the concern of referee#1 with respect to the different use of the tagging vs. perturbation methods we revised the Introduction and added a new Table, now numbered 1, which describes the different (scientific) questions the methods answer. Further, we clarified the used terminology throughout the text to avoid misunderstandings. In addition, we added also a short remark on this in the Conclusions.

We checked the language of the manuscript again and clarified some issues. In accordance with the comments from referee#2 we revised Figure 9 and 13 and adjusted the font sizes and spacings in all other figures slightly.

Attached are the replies to the two referees (original comments in italic, answers in normal fonts, changes in the revised manuscript in bold) together with the revised manuscript. In the revised manuscript all modifications are highlighted (latexdiff).

We are looking forward to your reply,

Mariano Mertens
(on behalf of all co-authors)

Dear referee#1
Thank you very much for your review of our revised version of the manuscript acp-2019-715. Please find our replies to your comments below. In the following, referee comments are given in italics, our replies in normal font, and text passages which we included in the text are in bold.

*This revised version of the paper has been very much improved although the English should yet be reviewed. In this respect, I listed again some possible improvements (see the minor comment section) but the whole text would need to be revised. I remain with one major question / concern that has not been answered by the Authors and I would appreciate some answers before I could recommend publication.*

We thank referee#1 very much for honouring our work on the revised manuscript. We hope that your last major concern is addressed with our changes (see below). Further, we would like to thank you for your suggestions regarding the English. We adapted most of them. Further, we would like to point out that the manuscript was read by several people which were not involved in the drafting process of the manuscript. However, we have checked the language again for the revised version.

*The major concern / question I would like to come back to is a point I raised in my first review. I believe it was not answered satisfactorily. I agree with the Authors that sensitivity analysis and tagging are two approaches that are used to answer dierent questions. I also agree that on one hand, sensitivity delivers impacts that are used to answer the policy question: what is the potential eect of an emission reduction on air quality levels? And on the other hand, tagging delivers contributions. But I yet do not understand to what policy or science question contributions answer. How should we use contributions? For what purpose? For example, the Authors state that their goal is to quantify contributions and that tagging is well suited to do this. I agree with this statement but what for? They also state that contributions are the rst step in understanding the inuence of dierent emission sources. But both impacts and contributions would serve that purpose. What is the added value of contributions to understand the influence of emission sources? As the entire work is devoted to the analysis and quantification of contributions, some information should be provided to clarify these points.*

Reply: We are sorry that you feel that this point has not been addressed sufficiently in our reply/revised version. Our goal is to analyse the share of ozone attributable to land transport emissions. This means that we want to fully decompose the ozone levels into the contributions of all emission sectors/other sources. This decomposition allows to answer fundamental scientific questions such as: What determines tropospheric ozone values? Indeed, this can be seen as an purely academic question, but it is an important question to understand the tropospheric ozone budget. We understand that policy makers might be more interested in the results of perturbation studies, which deliver impacts of potential emission reductions. However, this is not the focus of the current manuscript. We believed that these points were mentioned clearly in the introduction of the revised manuscript. This was obviously not the case. Therefore we revised parts of the introduction (see below) and added Table 1, which compares impact and contribution and the scientific questions they address directly.Moreover, we clarified the used terminology throughout the text and introduced the terms to avoid misunderstandings.

In addition, we would like to point out that the concept of analysing contributions to ozone levels is well established in the scientific community. The concept was used in the past in several scientific publications focusing on global as well as on regional scales. Examples are: Horowitz and Jacob (1999); Lelieveld and Dentener (2000); Meijer et al. (2000); Dunker et al. (2002); Grewe (2004); Sudo and Akimoto (2007); Butler et al. (2011); Emmons et al. (2012); Kwok et al. (2015); Butler et al. (2018); Pay et al. (2019); Lupaşcu and Butler (2019).

The new section in the Introduction reads:

**To quantify the influence of a specific emission source, such as land transport emissions on ozone, source apportionment methods are needed. Typically, two different methods are used for source apportionment. The first method is the *perturbation method*. In the perturbation method (also known as sensitivity analysis, brute-force, or zero-out) the results of two model simulations, one with all emissions and one with changed emissions, are compared. The second method is based on a labelling technique (known as tagging) to attribute specific pollutants, such as for instance ozone, to specific emission sources. Hereafter, we refer to this method as *source attribution*. As outlined in different studies (Wang et al., 2009; Grewe et al., 2010; Clappier et al., 2017) both methods answer different question because of their fundamentally different concepts. The perturbation method quantifies the change of ozone due to an emission change. In this method the sensitivity of ozone to this emission change is analysed based on a Taylor approximation (Grewe et al., 2010). In contrast, source attribution gives no information about the sensitivity of ozone to an emission change. Instead, the share of ozone which is caused by the emissions of a specific emission source for a given state of the atmosphere is quantified. Therefore, we use hereafter the terms 'impact' for the results of the perturbation method and 'contribution' for results of source attribution. The characteristics of impacts and contributions are listed in Table S1. By design, source attribution methods decompose the ozone budget completely into their respective contributions (this could be emission sectors, geographical regions, combinations of this or other measures). Contributions calculated by source attribution are of interest for academic purpose to study the tropospheric ozone budget and to increase scientific understanding about factors**

determining ozone levels (e.g. Horowitz and Jacob, 1999; Lelieveld and Dentener, 2000; Meijer et al., 2000; Dunker et al., 2002; Grewe, 2004; Sudo and Akimoto, 2007; Dahlmann et al., 2011; Butler et al., 2018). Further, the knowledge about contributions can help the planning of mitigation options by finding the emission source, which contributes most to ozone (e.g. Kwok et al., 2015; Valverde et al., 2016; Pay et al., 2019). Furthermore, the contributions are very valuable for assessing possible changes in the tropospheric ozone budget due to changes of emissions or climate. However, contributions provide no information about the sensitivity of ozone with respect to an emission change, such as the resulting ozone change, when emissions of a specific emission source become reduced or increased. The answers to such questions require the perturbation method, which quantifies the impact of an emissions change onto ozone. In contrast to the contributions, the effect of an emission reduction (and therefore the impact) can be measured. However, the results of the perturbation approach provide no information about how the effect of an emission reduction is altered by compensating effects of other emission sources (for instance an increase of ozone productivity of an unmitigated source). In order to assess such effects, perturbation and source attribution methods must be combined (see Mertens et al., 2018).

In addition, we added a paragraph to the conclusions:

As discussed in our introduction (see Table S1) contributions, provide the share of ozone caused by specific emissions. With respect to mitigation options it is important to point out that these contributions provide no information about how ozone changes if the corresponding emission sources are reduced. As discussed, this question can only be assessed with the perturbation approach. However, the contributions indicate how important different emission sources are for the tropospheric ozone budget in Europe.

*1) P25 L3-4 The Authors state that land transport contributions increase with higher ozone values, hence land transport emissions are an important driver of large ozone values. In my view we would get the same interpretation with low ozone values. In the extreme case of ozone titration, large transport contributions would increase with lower ozone values. Could the Authors comment on this?*

Reply: This is indeed an interesting question. On P25L3ff we present the results of our analysis (focusing on high ozone values) and discuss these results. From these results we cannot draw any conclusion with respect to low ozone values. However, we agree that within cities, where large emissions of NO take place from the transport sector, a large fraction of ozone might be caused by these transport emissions (even though ozone itself is low). However, we did not analyse this and therefore we don't want to speculate about it. Nevertheless,

this is a good idea for future work.

*2) What are the exact differences between sections 4 and 6? The section titles look very similar but the approaches followed in each section differs. I would suggest adding a few lines to clarify this.*

Reply: Thanks for the comment. We changed the title of Sect. 6 to 'Contribution of land transport emissions to net ozone production in Europe' to make this more clear. Further, we adapted the description at the end of Sect. 1.

*P31 L6-8: Stating that the contributions to ozone are more robust, i.e. less dependent on the background than impacts. I think this statement is confusing. The real world shows variations with respect to the background but contributions do not. Contributions are indeed more robust but they do not represent the real world. Could the Authors comment on this?*

Reply: We don't agree with the statement from referee#1 that contributions do not represent the 'real world'. For a given state of the atmosphere, contributions show the share of emission sources in the ozone budget. Clearly, contributions do not show how ozone would change if emissions are changed (if that is what the referee#1 means with real world). To clarify this, we changed the paragraph from:
"This is in line with previous work, stating that the contributions to ozone are more robust, i.e. less dependent on the background, as the perturbations or impacts (Grewe et al., 2012, 2019)."

to

**This is in line with previous work, stating that for ozone source attribution contributions instead of impacts should be used (Grewe et al., 2012, 2019).**

*4) P2 L28-29: In practice, the sensitivity or perturbation method is not restricted to small perturbations around the base case. Larger perturbations can be applied while keeping linear responses but this strongly depends on the time average selected and on the pollutant considered. For annual averages, reductions up to 50% have been tested and the model responses remain linear for PM but also for O3 (see e.g. Thunis et al. 2015). Obviously this is not the case for episodes, but even then reductions up to 20% can be applied. I would therefore adapt the text in the introduction to mention these points.*

We rephrased this point to:

**Only for small perturbations around the base state (w.r.t. the chemical regime) the response of ozone to a small emission change can be considered as almost linear. Whether a response to an emission**

change is nearly linear depends on the chemical regime, and therefore the region and the considered time period. Thus, the perturbation approach does not allow for a complete ozone source attribution (e.g. Wild et al., 2012), because the impacts calculated for the different sectors do not sum up to 100 %. This leads to an underestimation of the contribution of specific emission sources to ozone, if these impacts are used for source attribution. As an example, Emmons et al. (2012) reported that tagged ozone is 2–4 times larger than the impact calculated by the perturbation approach.

*P3 L2: This sentence is rather unclear as it seems to mix the concept of contributions with impacts.*
Reply: This sentence is removed from the revised Introduction.

*P3 L28/29: The sentence is grammatically incorrect and unclear. Please revise English*
Reply: We rephrased it to: **ozone and ozone precursors which are advected towards Europe (i.e. significantly influenced by boundary conditions of the regional model) are not attributed to specific emission sources (or regions) but are attributed to the boundary conditions only.**

*P4 L9: trough –> through but I would suggest inserting this point: choice of source... as an independent points* Reply: We added this as additional point which reads: **uncertainties due to the choice of source attribution methods**

*P5 L 31: an one –> a one* Reply: Fixed

*P7 L 21: As discussed* Reply: Changed.

*P8 L8-14: Again here, the Authors seem to mix the impact and contribution concepts (especially L8 and 9). Please clarify*
Reply: No, we are well aware of these two concepts. However, contributions can be combined with impacts, i.e. in all simulations (with all emissions and with modified emissions) the contributions can be calculated and used for additional analyses of the calculated impacts. To make this more clear we changed this sentence to:

**If the tagging scheme is used in addition to the perturbation approach (see Table S1) to investigate the influence of mitigation options, the approach of Grewe et al. (2017) leads to the effect that in VOC limited regions a $NO_x$ emission reduction of an emission sector reduces the contribution of that sector, and increases the contribution of the other sectors.**

*P9 L5: as the -> than the*
Reply: Fixed

*P9 L15/16: The sentence is grammatically incorrect and unclear. Please revise English*
Reply: We rephrased the sentence to:
**Finally, the land transport emissions are estimated by combining the activity data of the traffic simulations with corresponding emissions factors.**

*P9 L25: sectors anthropogenic non-traffic -> anthropogenic non-traffic sectors*
Reply: No. We mean the two sectors called anthropogenic non-traffic and shipping (see also Table 3).

*P10 L 4: and the emission differences between both simulations*
Reply: Rephrased it.

*P10 L14: simulations*
Reply: Fixed

*P11 L3: too less -> too low*
Reply: Fixed

*P12 L10: The sentence is grammatically incorrect and unclear. Please revise English*
Reply: Changed from:

'This leads to a around 1 percentage point lower contribution of anthropogenic emissions in COSMO-CLM/MESSy compared to EMAC.'

to:

**Altogether COSMO-CLM/MESSy simulates an approximately 1 percentage point lower contribution of anthropogenic emissions to ground-level ozone compared to EMAC (see Mertens et al., 2020).**

*P13 L4: at the European scale*
Reply: Fixed

*P13 L17: to the share*
Reply: Fixed

*P14 L1: these differences between contributions*
Reply: Fixed

*P18 L3: ground level ozone*

Reply: Fixed (but with ground-level).

*P18 L4: non-traffic is quite unclear: could we say what it is rather than what it is not?*
Reply: The tagging category is called anthropogenic non-traffic. The detailed definition is given in Table 3. At the current stage of the manuscript we do not want to change the names of the tagging categories anymore. However, we will discuss a different name for future work.

*P18 L18: DJF ground level O3*
Reply: Because of our experience with the copy editing of Copernicus we stick to ground-level. We changed the beginning of the sentence to:

**DJF ground-level $O_3^{tra}$ simulated by**

*P18 L19: absolute –> real and these contributions –> contributions reported in this work?*
Reply: We changed the sentence to:

**Even lower ground-level $O_3^{tra}$ is simulated near some hot-spots due to ozone titration.**

*P19 L11/L2: The sentence is unclear. Please revise*
Reply: We changed it to:

**The relative contribution of $O_3^{tra}$ in the 'inflow' region is about 9 % in both seasons and for both European emission inventories.**
*P24 L6 & 8: percentiles*
Reply: Fixed

*P24 L17: representations*
Reply: Fixed

*P24 L33 Please revise and clarify sentence*
Reply: We changed the sentence to:

The medians of the relative contribution of other anthropogenic emissions (i.e. the emission sectors anthropogenic non-traffic and aviation) range in all regions **from** 17 to 25 % (Fig. 11c and d).

*P28 L 11: differences of the –> differences between*
Reply: Fixed

*The sentence is unclear. Please revise*
Reply: We changed the sentence to:

$P_{O3}$ **of the anthropogenic non-traffic category differs by roughly 30 %**
**between REF and EVEU, whereas the total net ozone production**
**differs by roughly 15 %. Due to the lower total emissions in the VEU**
**emission inventory compared to the MAC inventory, less ozone is**
**produced in the former.**

*P30 L27-28: This also depends on the availability of VOC*
Reply: Of course. We changed the sentence as following to make this more clear:

This is caused by the non-linearitiy of the ozone chemistry, **including the**
**availability of VOCs**, and the strong interdependence of ozone production
and meteorological conditions (e.g. Monks et al., 2015).

*P31 L11-12: The wording sensitivity approach in this sentence is misleading as*
*this is the same term used to differentiate impacts and perturbations.*
Reply: Thanks for this comment. This is indeed misleading. We rephrased
it to: **As discussed, these two methods examine proxies of the ozone**
**sensitivity to determine, whether ozone production is** $NO_x$ **or VOC**
**limited.**

*P32 L2: their approach*

Reply: Fixed

*P32 L14: The sentence is grammatically incorrect and unclear. Please revise*
*English*

Reply: we rephrased the sentence to: **In general, the various studies dis-**
**cussed above do not provide sufficient information on the emissions**
**parametrisations and inventories used (e.g. for lightning-$NO_x$, soil**
$NO_x$ **and biogenic VOCs) to fully analyse these differences.**

*P32 L18: between the two emission*

Reply: Fixed

*P32 L31: by up to*

Reply: Fixed

*P33 L12-13: The sentence is grammatically incorrect and unclear. Please revise*
*English*

Reply: We changed the sentence to:

In addition, measurements of specific episodes or in specific regions (e.g. in plumes of cities, in regions with strong lightning activity or events of stratospheric intrusions) can help to assess the diagnosed contributions by investigating, if these contributions are in a plausible range. Further, the influence of model biases on the analysed contributions can be estimated, but a direct evaluation of these contributions is not possible.

Best regards,
Mariano Mertens
(on behalf of all co-authors)

*The authors have done a good job of answering the two reviews, resulting in a much-improved manuscript. Some of the figures have an unpolished look to them though. For example, it's difficult to make sense of the labels and arrows in Figure 9, and the text below the label bar of Figure 13 seems disproportionately large. It also seems that Figure 11(c) has some problem with the offset of the dividing vertical bars. My only remaining suggestion is that the authors polish their figures a bit before final publication.*

We thank referee#2 very much for honouring our work on the revised manuscript. Indeed, Fig.11 (c) had some problem which we fixed. Further, we revised Figure 9 and adapted the font-size and spacing in all figures slightly (including Fig.13).

Best regards,
Mariano Mertens
(on behalf of all co-authors)

[revised manuscript text omitted]